# Global prevalence and correlates of mpox vaccine acceptance and uptake: a systematic review and meta-analysis

Sahabi Kabir Sulaiman [1] ✉, Fatimah Isma'il Tsiga-Ahmed [2], Muhammad Sale Musa [1],
Bello Tijjani Makama[3], Abdulwahab Kabir Sulaiman [4,5] & Tijjani Bako Abdulaziz [6]

## Abstract

**Background** Vaccination has been recommended as one of the most potent ways of controlling the mpox (formerly, monkeypox) outbreak, particularly among high-risk groups. Here, we evaluated the prevalence of mpox vaccine acceptance and uptake globally.
**Methods** We searched multiple databases for peer-reviewed studies published in English from May 2022 to 25th November 2023 that evaluated mpox vaccine acceptance and/or uptake. We fit a random-effects model meta-analysis to calculate the pooled mpox vaccine acceptance and uptake rates, with their 95% confidence intervals (CI) across population outcomes. We performed subgroup analyses among the six World Health Organization (WHO) regions (Africa [AFR], Region of the Americas [AMR], South-East Asia Region [SEAR], European Region [EUR], Eastern Mediterranean Region [EMR], and the Western Pacific Region [WPR]), as well as among select population subgroups.
**Results** Of the 2531 studies screened, 61 studies, with a cumulative sample size of 263,857 participants from 87 countries were eligible for inclusion. The overall vaccine acceptance and uptake rates were 59.7% and 30.9% globally. Acceptance and uptake rates among the LGBTQI+ community were 73.6% vs 39.8% globally, 60.9% vs. 37.1% in AMR, 80.9% vs. 50.0% in EUR, and 75.2% vs. 33.5% in WPR. Among PLHIV, vaccine acceptance and uptake rates were 66.4% vs. 35.7% globally, 64.0% vs. 33.9% in AMR, 65.1% vs. 27.0% in EUR, and 69.5% vs. 46.6% in WPR. Among healthcare workers, vaccination intention was 51.0% globally.
**Conclusions** Tailored interventions are needed to bolster confidence in the mpox vaccine, maximize vaccine uptake, and increase vaccine access to close the gaps between acceptance and uptake especially among key populations residing in regions with low rates of acceptance and uptake.

## Plain language summary

Mpox is an infection caused by the monkeypox virus and is transmitted through direct contact with infected animals or people, or indirectly through contact with contaminated materials. An unprecedented mpox outbreak spanning all continents occurred in 2022. Vaccination against the infection by high-risk groups, including the LGBTQI+ community and frontline health-care workers has been recommended by the WHO as essential to outbreak control. To investigate the rates and factors associated with mpox vaccine acceptance and uptake across population subgroups (LGBTQI+ community, healthcare workers, people living with HIV, and the general public), we undertook this global systematic review and meta-analysis of the available evidence. Our results reveal substantial global and regional variations in the rates of mpox vaccine acceptance and uptake across population groups, with wide acceptance-uptake gaps, indicating the need for behavioral interventions to increase mpox vaccine confidence and uptake.

The World Health Organization (WHO) has recommended vaccination as one of the most effective interventions required to prevent and control the unprecedented spread of mpox (formerly monkeypox), a zoonotic viral infection that has infected about 92, 783 individuals across116 countries, with 171 deaths reported as of November 2023[1–3]. Successful control of the outbreak requires optimal acceptance and uptake of the vaccination against mpox, particularly among those designated as high-risk groups, such as the

lesbian, gay, bisexual, transgender, queer, and intersex+ (LGBTQI+) community[4]. However, this outbreak occurred at a time when the world is witnessing all-time high levels of vaccine hesitancy[5–8], defined by the WHO as "a delay in the acceptance or refusal of vaccination despite the availability of vaccination services"[9]

Vaccination against mpox may be provided to individuals at high risk of the infection as primary preventive vaccination (PPV) prior to exposure

[1]Department of Medicine, Yobe State University Teaching Hospital, Damaturu, Nigeria. [2]Department of Community Medicine, Bayero University Kano/Aminu Kano Teaching Hospital, Kano, Nigeria. [3]St Helens and Knowsley Teaching Hospital, NHS Trust, Prescot, UK. [4]Department of Medicine, Murtala Muhammad Specialist Hospital, Kano, Nigeria. [5]Kwanar Dawaki COVID-19 Isolation Center, Kano, Nigeria. [6]Department of Neurosurgery, Houston Methodist, Houston, TX, USA.
✉e-mail: sahabikabir25@gmail.com

to the mpox virus, or as post-exposure preventive vaccination (PEPV) for contacts of mpox cases[4,10]. In addition to the previously employed smallpox vaccine, which has been shown to be highly effective in protecting against mpox[11], newer vaccines, including the MVA-BN, LC16, and the ACAM2000 have been approved in many countries for the prevention of mpox[10,12]. A recently published systematic review shows that these vaccines are highly effective, safe, and immunogenic, depending on the number of doses administered and that vaccines against smallpox offer cross-protection against mpox[13]. However, in people living with HIV (PLHIV), a population accounting for about four in ten confirmed mpox cases[14–18], safety concerns have precluded the use of the ACAM2000[12]. In a recently published global case series reported a severe form of mpox resembling an AIDS-defining condition, with a mortality rate that is as high as 25% among people with advanced HIV[17].

Previous systematic reviews[19–21] have attempted to identify key determinants of intention and hesitancy to vaccinate against mpox. However, these reviews were limited by having a small number of included studies and lacking representation across all six WHO regions. Also, none of these reviews reported the regional rates of intention to vaccinate against mpox among key populations designated as high-risk groups by the WHO, including the LGBTQI+ community and healthcare workers. Furthermore, the previous meta-analyses did not report vaccine uptake rates or the global vaccine acceptance rate among vulnerable groups, such as PLHIV.

In view of these literature gaps, we conduct this systematic review and meta-analysis of 61 studies involving 263,857 participants to report the global and regional prevalence of mpox vaccine acceptance and uptake among various populations, including the LGBTQI+ community, PLHIV, healthcare workers and the general public as well as the pooled rates of uptake among people who indicated their willingness to be vaccinated. Our findings reveal substantial global and regional variations in the acceptance and uptake rates of the mpox vaccine across these population groups. Moreover, we also find wide a acceptance-uptake gaps of the mpox vaccine, including across key populations like the LGBTQI+ community. These findings call for deliberate efforts to increase access to mpox vaccine especially for at-risk group in order to close the gap between intention to vaccinate and the actual vaccine uptake.

## Methods

The present review followed the Preferred Reporting Items for Systematic Reviews and Meta-Analyses (PRISMA) guidelines[22]. The protocol for this review was registered with the International Prospective Register for Systematic Review (PROSPERO ID: CRD42022378564).

The initial literature search was conducted between 15th to 25th February 2023 in multiple electronic databases, including Medline, Embase, PubMed, Google Scholar, Web of Science, Scopus, and PsycINFO to identify studies evaluating the acceptance and/or uptake of the mpox vaccine. An updated literature search was performed on 25th November 2023. A detailed search strategy, comprising key terms, the Boolean operators, 'AND' and 'OR,' and Medical Subject Headings (MeSH), was developed for PubMed and adapted for the other databases (Supplementary Data 1). Briefly, the key terms used included "Monkeypox," "Mpox," "Vaccine," "Vaccination," "Uptake," "Acceptance," "Willingness," "Intention," "Access," "Hesitancy," "refusal," "Uncertainty," "Indecision," "Determinants," "Factors," "Correlates," and "Predictors".

### Criteria for study inclusion/exclusion

To include studies reporting the rates oof mpox vaccine acceptance, intention/willingness and uptake, we followed the guideline of the CoCoPop (condition, context, population)[23] statement for review of studies reporting on prevalence/incidence. Accordingly, we included original full-text articles reporting any of this study's outcomes of interest: prevalence rates of intention to vaccinate against mpox, prevalence of vaccine uptake, or factors associated with mpox vaccine acceptance or uptake.

We excluded studies that: (1) were available only as abstracts or pre-prints; (2) evaluated conditional acceptance only (e.g. willingness to pay for

vaccination); (3) did not report on any of our primary outcomes of interest; (4) used only continuous variables to measure the outcome of interest without reporting the exact prevalence of acceptance, intention or uptake; and (5) involved only clinical trial of the mpox vaccine among participants without reporting the prevalence of vaccine acceptance, intention or uptake; (6) was conducted before the 2022 global outbreak.

### Study selection and eligibility

The initial literature search identified a total of 748 studies. After the removal of duplicate studies, a total of 309 articles were further screened independently using the Rayyan QCRI (Qatar Computing Research Institute)[24], based on title, then abstracts, and subsequently full-text by two investigators (SKS and MSM). All discrepancies were referred to and resolved by two senior authors (FIT and ATB). Following the screening process, 142 articles were found eligible, and following the application of our exclusion criteria, a total of 39 studies were included in the first stage of our literature search. (Fig. 1). We updated our literature search to include studies published by 25th November 2023. We, in addition, searched the Regional Office for Africa Library, the African Index Medicus, and the WHO Institutional Repository for Information Sharing to identify relevant studies. We also employed direct manual search and forward and backward citation tracking to retrieve studies. These cumulative searches yielded a total of 1783 studies from which we excluded 1,658 studies for duplication and other eligibility reasons. The full texts of the remaining 125 studies were retrieved and assessed for eligibility. Among these 125 studies, we excluded 2 preprint articles, 9 review articles, 39 studies already included in our initial literature search, and 74 studies that did not meet other inclusion/exclusion criteria. Consequently, our updated search identified 22 additional studies eligible for inclusion. Thus, a total of 61 studies were included in this review.

### Study outcomes

The primary outcomes were: 1. Intention, defined as the unconditional willingness to receive free mpox vaccine; 2. Uptake, defined as the actual receipt of one or more of any of the vaccines approved for mpox prevention and/or treatment during the 2022 outbreak; 3. Acceptance, which comprises both the intention to receive the mpox vaccine and the actual uptake of the vaccine, in line with previous studies[25]. Thus, in studies that only reported intention to accept the mpox vaccine among unvaccinated people, the prevalence of acceptance was defined as the proportion of individuals intending to be vaccinated among the entire cohort. However, for studies that reported both vaccine uptake and intention to vaccinate, the prevalence of acceptance was calculated as the proportion of those already vaccinated and those willing to be vaccinated among the study cohort.

We performed stratified analyses of the intention, uptake, and acceptance rate of the mpox vaccine for the overall populations and across the various population subgroups (such as the LGBTQI+ community, PLHIV, healthcare workers, and the general public) according to WHO global regions. We also evaluated the prevalence of vaccine uptake among people who indicated acceptance. Furthermore, we assessed the prevalence of vaccine intention and acceptance among PLHIV, as well as the factors associated with mpox vaccine acceptance, intention, and/or uptake.

### Data extraction

The Zotero software (version 6.0.15) was used for the retrieval of references as well as the removal of duplicate articles generated from the literature search. Thereafter, one investigator (SKS) used the Joanna Briggs Institute (JBI)[26] data extraction form to extract relevant data from the retrieved articles. The data extracted were rechecked by two other investigators (MSM and BTM). The information extracted from the included articles comprised the first author's name; publication year; study title; country/countries of study; study setting; study design sampling method; means of study administration; study period; publication status; sample size; the number of male participants; the mean/median age of the participants; the number of participants who indicated an intention to accept the mpox vaccine, actual uptake, and/or acceptance; the number of PLHIV participating in the study,

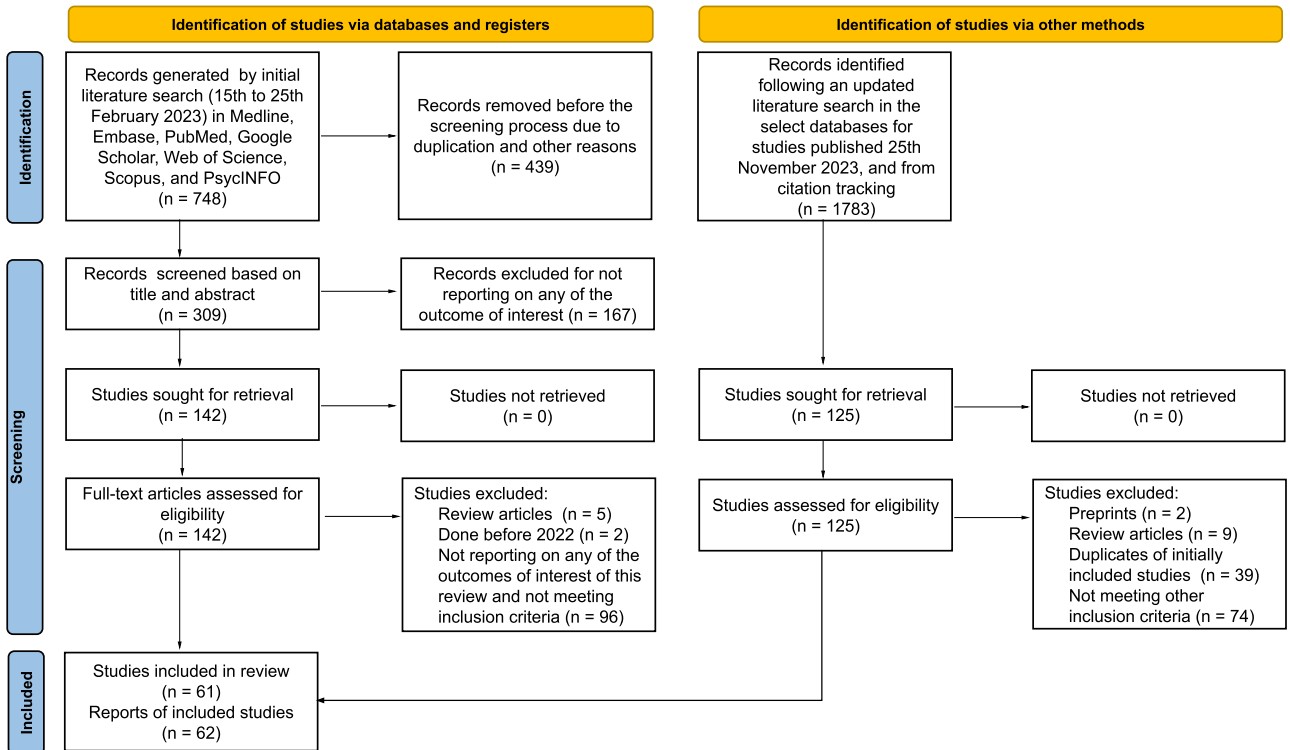

**Fig. 1 | PRISMA flow diagram of the included studies.** From a total of 748 studies retrieved from multiple databases during the initial literature search between the 15th and 25th of February 2023, we removed 439 studies due to duplication and many other reasons. We then screened 309 studies based on their titles and abstracts and excluded 167 studies for not reporting on any of the outcomes of interest. The remaining 142 studies were sought for retrieval and assessed for full-text eligibility, Subsequently, we removed: 5 review articles, 2 studies qualified by outcome measure but conducted before 2022 (on the basis of peer review process recommendation), and 96 studies for not reporting on any of the outcomes of interest and not meeting our other inclusion criteria. Therefore, a total of 39 studies having 40 reports were included in our initial search. At the request of the editorial and peer review process, we extended our literature search to studies published by 25th November 2023, repeated the search in the select databases, manual search and employed forward and backward citation tracking, generating 1783 studies. From this (1783), we excluded 1658 studies for duplication and other eligibility reasons and sought the remaining 125 for retrieval. From this (125), we excluded: 2 studies for being preprints, 9 studies for being reviews, 39 studies for being the initially included studies, and 74 studies for not meeting the inclusion criteria thereby making the remaining 22 studies from the update to qualify for inclusion. Therefore, a total of 61 studies (39 from initial and 22 from final literature searches, respectively) were finally included. We only included studies if they: (1) are original full-text publications; (2) are not reporting a conditional acceptance; (3) reported any of our outcomes of interest (acceptance, intention, uptake, and/or associated factors), (4) did not solely use a continuous variable to measure the outcome of interest; (5) were not clinical trials only; (6) were not conducted before 2022. *From:* Page MJ, McKenzie JE, Bossuyt PM, Boutron I, Hoffmann TC, Mulrow CD, et al. The PRISMA 2020 statement: an updated guideline for reporting systematic reviews. BMJ 2021;372:n71. doi: 10.1136/bmj.n71.

along with the number of PLHIV indicating their intention to accept the vaccine; and the factors independently associated with vaccine acceptability and/or uptake. The data extracted and used for this work is available at https://doi.org/10.17605/OSF.IO/FS5QH.

**Critical appraisal (quality assessment) of included studies**
Two investigators (SKS and MSM) independently reviewed all articles to critically appraise their methodological rigor using an adapted version of the Newcastle Ottawa Scale (NOS)[27] for cross-sectional studies, and the NOS for cohort studies[28]. The scale consists of seven items divided into three (3) major domains; 1) Selection, having a total of four items and a maximum score of five; 2) Comparability, having only one item and a maximum score of two; and 3) Outcome, having two items and a maximum score of three. Accordingly, a study is rated as having low (1–4), moderate (5–7), or high (8–9) quality of evidence. The scores of the two investigators were compared and reviewed by two senior authors (FIT and ATB), and where disputes occurred, a final consensus score was decided by the senior authors through revision and discussion of the articles. (Supplementary Data 2 and Supplementary Table 1).

**Statistical analysis**
The meta-analysis was performed using the metaprop command in Stata Version 15IC (StataCorp, College Station, Texas USA)[29]. Inverse variance weights were used to estimate the pooled prevalence rates of the mpox vaccine intention, uptake, and acceptance from the studies. Forest plots were used to present the results of the respective pooled proportions. The $I^2$ measure was used to assess the percentage of total heterogeneity (variation) across the studies. Accordingly, heterogeneity was categorized as low (0–25%), moderate (26–75%), and substantial (76% to 100%)[30]. Because the studies were highly heterogeneous, only the random effects model was used to illustrate the pooled proportions of the mpox vaccine intention, uptake, and acceptance, as recommended[31].

Stratified analyses were performed based on region and study population. To prevent the exclusion of some studies having proportions that are close to or equal to 1, the Freeman-Tukey double arcsine transformation was used[32–34]. The pooled proportions and weighted mean differences with their 95% confidence intervals (CI) were reported, and a *p*-value of <0.05 was considered significant.

To compute the prevalence of intention, we divided the number of those who were willing to be vaccinated by the total number of unvaccinated participants in the study and multiplied the proportion by a hundred. Furthermore, we computed the prevalence of vaccine uptake as the number of vaccinated participants divided by the total study participants multiplied by a hundred. The prevalence of acceptance was calculated as the number of participants in the acceptance group (intention + uptake) divided by the total sample size of the study multiplied by a hundred. To derive the

prevalence of uptake among participants that indicated acceptance, we divided the number of vaccinated participants by the total number of participants who indicated acceptance of the vaccine and multiplied that by a hundred.

To evaluate a potential effect of each included study on the prevalence estimates, we performed a series of leave-one-out sensitivity analyses among the overall study population and across population subgroups. This analysis involved the iterative removal of a single study to report the pooled estimated prevalence rates without the excluded study. This process was repeated until all studies were individually excluded.

To evaluate potential publication bias among the included studies, we used funnel plots[35], and Egger's test[36], with a P-value > 0.05 indicating no statistically significant evidence of publication bias. We also evaluated publication bias by assessing for asymmetry in the Doi plot, a plot of the normal-quantile versus effect size using the LFK index[37,38]. An LFK index beyond ±1 was deemed to be consistent with Doi plot asymmetry. Where Doi plot asymmetry is observed, we sequentially removed (trimmed) studies potentially causing the asymmetry until Doi plot symmetry is achieved, borrowing from Duval and Tweedie's (2000) Trim and Fill method. We then compared the outcome prevalence before and after trimming of studies to assess the effect of the studies causing Doi plot asymmetry on the prevalence estimates.

### Reporting summary
Further information on research design is available in the Nature Portfolio Reporting Summary linked to this article.

## Results
### Characteristics of the studies
The present systematic review included 61 peer-reviewed studies with a cumulative sample size of 263,857. The prevalence of intention to vaccinate was not reported by nine studies[39–47], uptake rates were reported by seventeen studies[39–42,44–56], rates of uptake among those who indicated acceptance were reported by eight studies[48,50–56], and mpox acceptance rate was reported by all included studies. Overall, all six WHO regions were represented, with AFR (Algeria, Ghana, Nigeria) having four studies[39,57–59], AMR having eleven studies[40,42,45,47,54,55,60–64], EMR (Iraq, Pakistan, Saudi Arabia) having eight studies[65–72], EUR (all WHO European member countries) having seventeen studies[41,44,48,51,52,56,73–83], SEAR (Indonesia, Bangladesh, Japan) three studies[84–86], and the WPR (China, Malaysia, Australia) having twelve studies[49,50,53,87–95]. The acceptance rate among PLHIV was reported by fourteen studies[41,47,49,50,52–54,63,74,78,79,87,91,96], the rate of intention to be vaccinated was reported by nine[50,52,63,74,78,79,87,91,96], while uptake rate was reported by eight studies[41,47,49,50,52–54,96]. Among the LGBTQI+ community, the acceptance rate was reported by twenty-one studies[42,45,47,49–53,55,56,62,63,74,78–80,87,89,91,93,95], rate of intention to vaccinate was reported by seventeen stuides[50–53,55,56,62,63,74,78–80,87,89,91,93,95], while rate of uptake was reported by studies[42,45,47,49–53,55,56]. The rate of intention to vaccinate among healthcare workers was reported by fifteen studies[57,59,61,66,69,72,73,75,82–84,92,94,97]. Among the general public, the acceptance rate was reported by nineteen studies[39–41,44,48,51,58,60,64,65,67,68,71,81,85,86,88,93,98], rate of intention to vaccinate was reported by fifteen stuides[48,51,59,60,64,65,67,68,71,81,85,86,88,93,98], while rate of uptake was reported by six studies[39–41,44,48,51]. The rate of intention to vaccinate among university students was reported by three studies[70,89,99].

Among all included studies, two were multiregional, having involved participants from more than one WHO region[97,98]. Apart from these two multiregional studies. All other studies were single-country studies, with US having the highest number of studies (nine)[40,42,43,45,47,54,60,61,96] followed by the China (eight)[49–51,65,67,75,89] and Saudi Arabia (four)[65–67,69]. In all, a total of 87 countries were represented in this review. All studies were conducted in 2022 and used non-probability sampling techniques for participant selection. Except for four studies that were utilized prospective cohort design[41,43,44,46], all studies were cross-sectional. Also, five studies used electronic record for data collection[40,41,43,46,49], while all the other studies were

online surveys. The largest sample size per study was 119,345[43], while the smallest was 75[84]. The study-level determinants of the mpox vaccine acceptance were reported in 42 out of the 60 studies included in this review (Supplementary Data 3).

Due to a substantial overlap of participants in the study by Zucker et al. (N = 2025 participants)[46] with that of Sagy et al. (N = 2054 participants)[41], we used only one of the studies (Sagy et al.) for quantitative synthesis. The study by Salih et al.[43] did not provided total number of uptake and n0n-uptake and was also not included for quantitative synthesis. Therefore, bringing the total number of studies eligible for meta-analysis to 59.

### Risk of publication bias
Among the studies reporting overall acceptance of the mpox vaccine (n = 59), visual inspection of Begg's funnel plot showed symmetry, and Egger's test similarly did not demonstrate evidence of publication bias (p = 0.856) (Supplementary Fig. 1). No evidence of publication bias was demonstrated among studies (n = 51) reporting overall intention to vaccinate against mpox by both Begg's test and Egger's test (p = 0.993) (Supplementary Fig. 2). For studies reporting overall uptake rates of the mpox vaccine (n = 17), evidence of publication bias was demonstrated by Begg's test and Egger's test (p = 0.022) (Supplementary Fig. 3). However, we did not found evidence of publication bias among studies (n = 9) reporting rates of mpox vaccine uptake among people who indicated acceptance using both Begg's test and Egger's test (p = 0.156) (Supplementary Fig. 4).

The Begg's test and Egger's test (p = 0.168) performed to check for the evidence of publication bias for studies (n = 14) reporting acceptance rate in PLHIV did not show evidence suggesting publication bias (p = 0.168) (Supplementary Fig 5). Similarly, no evidence suggesting publication bias was found according to the findings of our Begg's test and Egger's test (p = 0.215) for studies (n = 9) that reported the rate of intention to vaccinate among PLHIV (Supplementary Fig 6). Also, our assessment did not reveal evidence to suggest publication bias for the 9 studies reporting the rates of uptake among PLHIV by both Begg's test and Egger's test (p = 0.165) (Supplementary Fig 7).

Assessment of publication bias using Begg's test and Egger's test (p = 0.060) for the 21 studies reporting rate of acceptance among the LGBTQI+ community similarly revealed no evidence (Supplementary Fig. 8). There is also an absence of publication bias evidence among studies (n = 17) reporting the rate of intention to vaccinate against the mpox among the LGBTQI+ community based on the findings of our Begg's test and Egger's test (p = 0.202) (Supplementary Fig. 9). Our analysis of the 10 studies reporting uptake rates of the mpox vaccine among the LGBTQI+ community using the Begg's test and Egger's test (p = 0.081) did not show evidence of publication bias (Supplementary Fig. 10).

Publication bias was not observed among the 15 studies reporting the rate of intention to vaccinate against the mpox among healthcare workers from Begg's test and Egger's test (p = 0.443) (Supplementary Fig. 11). For the studies (n = 19) reporting acceptance rates among the general public, evidence of publication bias was observed by both Begg's test and Egger's test (p = 0.001) (Supplementary Fig. 12). Evidence of the presence of publication bias was also observed by both Begg's test and Egger's test (p = 0.003) among the 15 studies reporting rates of intention to vaccinate among the general public (Supplementary Fig. 13). However, no evidence of publication bias was observed among studies (n = 6) reporting the rate of intention to vaccinate against the mpox among the general public from the Begg's test as well as Egger's test (p = 0.473) (Supplementary Fig. 14).

Furthermore, Doi plot asymmetry was observed in the meta-analysis of studies reporting overall global prevalence of acceptance, uptake, and intention (Supplementary Fig 15–28). However, after trimming of studies potentially causing the Doi plot asymmetry, the prevalence estimates for acceptance, uptake, and intention only changed by 0.2, 1.3, and 1.3, respectively (Supplementary Table 4). Conversely, Doi plot asymmetry was not observed in the meta-analysis of studies reporting vaccine uptake among those who indicated their willingness to receive the vaccine. The Doi plots and LFK index, as well as the sensitivity analysis, for the stratified analysis

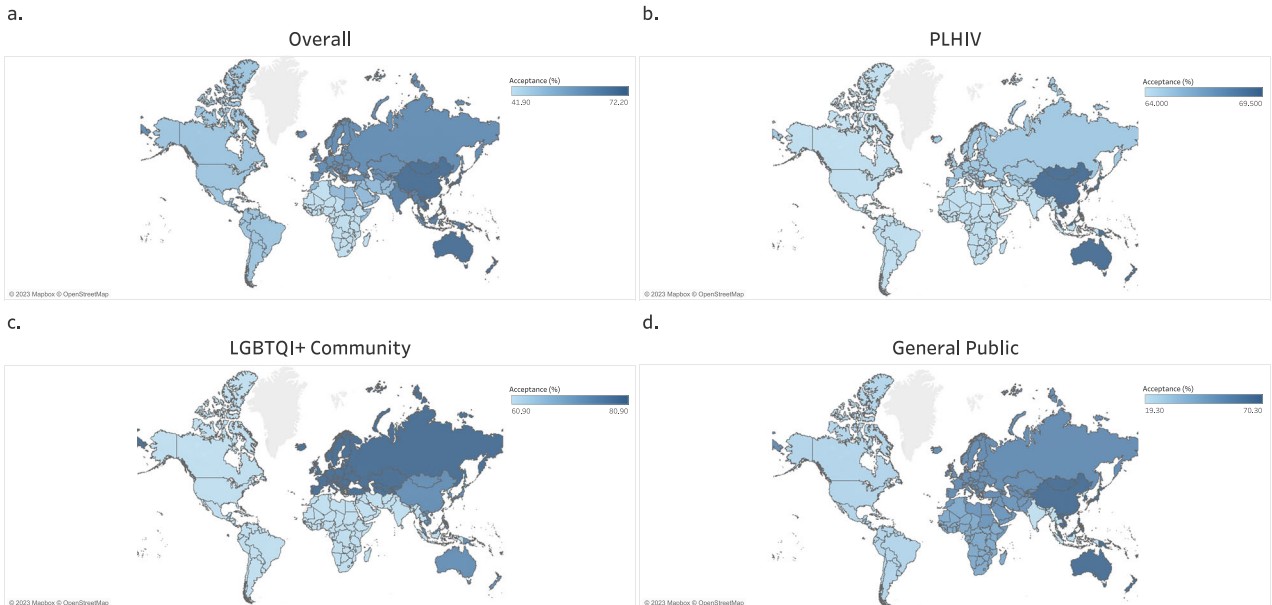

**Fig. 2 | Global map showing the regional pooled prevalence of mpox vaccine acceptance across four population groups (overall, PLHIV, LGBTQI+ community, and the general public). a** Population-wide (overall) acceptance rates of the mpox vaccine across all six WHO global regions. **b** Acceptance rates of the mpox vaccine among PLHIV according to WHO global regions (using data available for only three regions: AMR, EUR, and WPR). **c** Acceptance rates of the mpox vaccine acceptance among the LGBTQI+ community according to WHO global regions (using data available for only three regions: AMR, EUR, and WPR). **d** Acceptance rates of the mpox vaccine among the general public according to all six WHO global regions. Darker areas indicate higher rates, and lighter areas indicate lower rates. Maps adapted from OpenStreetMap under a Creative Commons licence CC BY-SA 2.0.

among various population subgroups are provided in Supplementary Fig 15–28 and Supplementary Table 2.

### Global and regional prevalence of mpox vaccine acceptance across population groups

The overall global prevalence of mpox vaccine acceptance, pooled from fifty-nine studies ($n = 142,487$ participants) included in the meta-analysis was 59.7% (95% CI, 51.1–68.1%) (Supplementary Data 4). Stratified by the WHO region, the SEAR had the highest acceptance rate at 72.2% (95% CI, 60.7–82.4%), followed by the WPR at 67.3% (95% CI, 5.7–100%), then EUR at 63.8% (95% CI, 54.6–72.6%), then EMR at 52.0% (95% CI, 44.2–59.8%), then AMR at 48.9% (95% CI, 24.9–73.2%), and AFR at 41.9% (95% CI, 38.5–45.3%) (Supplementary Data 4 and Fig. 2a). The pooled global acceptance rate among multiregional studies was 60.2% (95% CI, 41.1–77.8%) (Supplementary Data 4).

Based on data pooled from fourteen studies ($n = 7593$ participants), the global prevalence of mpox vaccine acceptance in PLHIV was 66.4% (95% CI, 51.4–79.9%) (Supplementary Data 4). According to the WHO region, the rate of mpox vaccine acceptance was 69.5% (95% CI, 45.3–89.1%) among PLHIV in the WPR, 66.4% (95% CI, 51.4–79.9%) among PLHIV in the AMR, and 65.3% (95% CI, 34.9–89.8%) among PLHIV in EUR (Supplementary Data 4 and Fig. 2b).

The prevalence of mpox vaccine acceptance based on data pooled from twenty-one studies conducted among 63,538 LGBTQI+ community members was 73.6% (95% CI, 67.2–79.6%) globally (Supplementary Data 4). This acceptance rate varied according to the WHO region, with the highest rate being among the LGBTQI+ community in the EUR at 80.9% (95% CI, 75.1–86.0%), followed by the WPR at 75.2% (95% CI, 60.2–87.6%), and then the AMR at 60.9% (95% CI, 35.2–83.8%) (Supplementary Data 4 and Fig. 2c).

The estimated global prevalence of mpox vaccine acceptance pooled from nineteen studies in the general public ($n = 56,518$ participants) was 50.9% (95% CI, 39.2–62.5%) (Supplementary Data 4). Stratified by the WHO region, the acceptance rate in the general public was 70.3% (95% CI, 68.6–72.0%) in the WPR, 56.5% (95% CI, 39.9–72.4%) in the EUR, 51.9% (95% CI, 46.0–57.8%) in the EMR, 43.3% (95% CI, 40.7–45.9%) in the AFR,

24.1% (95% CI, 7.9–45.7%) in the AMR, and 19.3% (95% CI, 18.8–19.7%) in the SEAR (Supplementary Data 4 and Fig. 2d). The pooled acceptance rate in the general public from a multiregional study was 48.9% (95% CI, 47.3–50.5%) (Supplementary Data 4).

### Global and regional prevalence of intention to vaccinate against mpox across population groups

The overall global rate of intention to vaccinate against mpox, pooled from fifty-one studies ($n = 127,359$ participants) was 60.9% (95% CI, 52.1–69.3%) (Supplementary Data 4). There was high variation in this intention rates across the six WHO regions, with the rate of intention to vaccinate being highest in the WPR at 73.5% (95% CI, 63.0–82.9%), followed by the SEAR at 67.3% (95% CI, 5.7–100%), then AMR at 59.5% (95% CI, 37.9–79.4%), then EUR at 59.3% (95% CI, 49.3–69.0%), then the EMR at 52.0% (95% CI, 44.2–59.8%) and AFR at 41.9% (95% CI, 36.6–47.4%) (Supplementary Data 4 and Fig. 3a). The pooled overall global rate of intention to vaccinate against mpox from multiregional studies was 60.2% (95% CI, 41.1–77.8%) (Supplementary Data 4).

Data pooled from nine studies in 6784 PLHIV puts the estimated the global prevalence of intention to vaccinate against mpox in this group at 75.0% (95% CI, 61.7–86.3%) (Supplementary Data 4). The rate of intention to vaccinate was 79.5% (95% CI, 50.2–97.7%) among PLHIV in the WPR, 74.8% (95% CI, 69.3–80.0%) among PLHIV in the AMR, and 71.7% (95% CI, 46.0–91.5%) among PLHIV in the EUR (Supplementary Data 4 and Fig. 3b).

Among the LGBTQI+ community, the estimated global prevalence of intention to vaccinate against mpox (pooled from seventeen studies in 61,118 participants) was 77.1% (95% CI, 72.3–81.5%) (Supplementary Data 4). According to the WHO region, the prevalence rate of intention to vaccinate was highest among the LGBTQI+ community in the AMR at 82.4% (95% CI, 76.0–88.1%), followed by the WPR at 78.0% (95% CI, 66.5–87.7%), and then the EUR at 74.2% (95% CI, 67.5–80.3%) (Supplementary Data 4 and Fig. 3c).

The global prevalence of intention to vaccinate against mpox pooled using data from fifteen studies in 9,064 healthcare workers was 51.0%

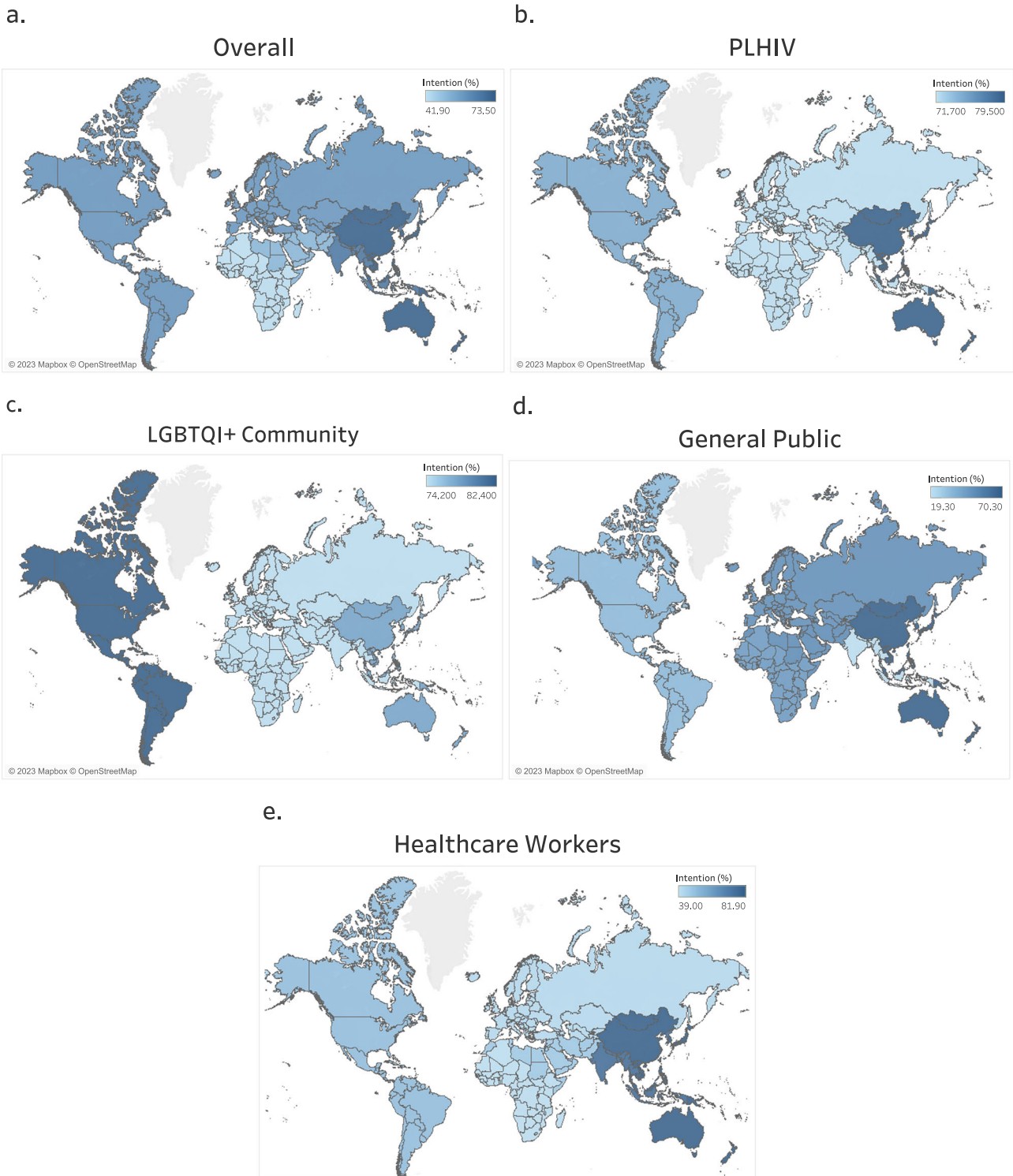

**Fig. 3 | Global map showing the regional pooled prevalence of intention to vaccinate against mpox vaccination across five population groups (overall, PLHIV, LGBTQI+ community, healthcare workers, and the general public).** **a** Population-wide (overall) rates of intention to vaccinate against mpox across all six WHO global regions. **b** Rates of intention to vaccinate against mpox among PLHIV according to WHO global regions (using data available for only three regions: AMR, EUR, and WPR). **c** Rates of intention to vaccinate against mpox among the LGBTQI+ community according to WHO global regions (using data available for only three regions: AMR, EUR, and WPR). **d** Rates of intention to vaccinate against mpox among healthcare workers according to all six WHO global regions. **e** Rates of intention to vaccinate against mpox among the general public according to all six WHO global regions. Darker areas indicate higher rates, and lighter areas indicate lower rates. Maps adapted from OpenStreetMap under a Creative Commons licence CC BY-SA 2.0.

(95% CI, 39.7–62.3 (Supplementary Data 4). Wide variations existed in the rates of healthcare worker intention to vaccinate against mpox across all six WHO region, with the highest rate of intention to vaccinate being 81.9% (95 CI, 80.0–83.7%) among healthcare workers in the WPR, then 77.3% (95% CI, 667–853.5%) in the SEAR, 49.7% (95% CI, 42.8–56.7%) in the AMR, 46.5% (95% CI, 28.9–64.6%) in the EMR, 40.8% (95% CI, 19.6–63.9%) in the EUR, and 39.0% (95% CI, 34.7–43.3%) in the AFR (Supplementary Data 4 and Fig. 3d). The pooled rate of intention to vaccinate among healthcare workers from a multiregional study was 54.5% (95% CI, 52.9–56.1%) (Supplementary Data 4).

Based in data available form fifteen studies (n = 43,810 participants), the estimated global prevalence of intention to vaccinate against mpox in the general public was 52. 3% (95% CI, 38.1–66.4%) (Supplementary Data 4). However, the results showed high variation according to the WHO region, with a highest rate the rate of 70.3% (95% CI, 68.6–72.0%) in the WPR, 51.9% (95% CI, 46.0–57.8%) in the EMR, 50.0% (95% CI, 25.3–74.8%) in EUR, 46.1% (95% CI, 42.2–50.1%) in the AFR, 33.9% (95% CI, 31.4–36.4%) in the AMR, and 19.3% (95% CI, 18.8–19.7%) in the SEAR (Supplementary Data 4 and Fig. 3e). The pooled rate of intention to vaccinate from a multiregional study was 48.9% (95% CI, 47.3–50.5%) (Supplementary Data 4). Furthermore, the global prevalence of intention to vaccinate against mpox among university students, pooled from three studies (n = 13,094 participants), was 59.4% (95% CI, 41.8–75.9%) (Supplementary Data 4).

## Global and regional prevalence of mpox uptake across population groups

Overall, the global prevalence of mpox vaccine uptake, pooled from seventeen studies (n = 26,186 participants), was 30.9% (95% CI, 21.0–41.7%) (Supplementary Data 4). Uptake rates varied by WHO region: 36.9% (95% CI, 15.4–61.6%) in EUR, 33.5% (95% CI, 21.9–46.3%) in the WPR, 28.3% (95% CI, 15.9–42.7%) in AMR, and 5.0% (95% CI: 3.7–6.7%) in AFR (Supplementary Data 4 and Fig. 4a).

The pooled global prevalence of mpox vaccine uptake among those who indicated their intention to receive the vaccine (uptake rate among the accepting group) based on data from nine studies involving 11,058 participants was 36.1% (95% CI, 19.9–54.1%) (Supplementary Data 4). Based on the WHO region, the uptake rate among the accepting group was 46.8% (95% CI, 42.1–51.5%) in the AMR, 33.7% (95% CI, 9.7–63.4%) in EUR, and 30.3% (95% CI, 28.4–32.1%) in the WPR (Supplementary Data 4 and Fig. 4b).

Globally, the prevalence of mpox vaccine uptake pooled from eight studies involving 1,933 PLHIV was 35.7% (95% CI, 27.3–44.6%) (Supplementary Data 4). PLHIV living in the WHO WPR region had the highest uptake rate at 46.6% (95% CI, 41.0–52.2%), followed by the PLHIV in the AMR at 33.9% (95% CI, 17.1–53.0%), and then EUR at 27.0% (95% CI, 24.7–29.4%) (Supplementary Data 4 and Fig. 4c).

The global prevalence of mpox vaccine uptake in the LGBTQI+ community pooled from ten studies (N = 8803 participants) was 39.8% (95% CI, 30.7–49.3%) (Supplementary Data 4). Stratified by the WHO region, the rate of uptake was highest in the EUR at 80.50% (95% CI, 24.4–75.6%), followed by the AMR at 37.1% (95% CI, 22.6–53.0%), and then the WPR at 33.5% (95% CI, 21.9–46.3%) (Supplementary Data 4 and Fig. 4d).

Among the general public, the estimated global prevalence of mpox vaccine uptake pooled from six studies involving 17,110 participants was 20.2% (95% CI, 6.9–38.3%) (Supplementary Data 4). The rate of uptake was 27.6% (95% CI, 3.1–64.0%) in the EUR, 12.8% (95% CI, 12.2–13.5%) in the AMR, and 5.0% (95% CI, 3.7–6.7%) in the AFR (Supplementary Data 4 and Fig. 4e).

The results of all the meta-analysis performed for the prevalence of mpox vaccine acceptance, intention, and uptake across all the population groups have been provided in the Supplementary Information (Supplementary Fig. 29–57).

## Sensitivity analysis

The overall population-wide pooled prevalence of mpox vaccine acceptance was not influenced by a single study according to our leave-one-out sensitivity analysis with the pooled estimates varying between 59.0% (95% CI, 53.0–65.0%; p = 0.000) and 61.0% (95% CI: 54.0–67.0%; p = 0.000) (Supplementary Fig. 58). Similarly, we found no evidence of an overriding influence of a single study on the pooled acceptance rates in our leave-one-out sensitivity analysis among PLHIV (64.0% (95% CI, 50.0–77.0%; p = 0.000) to 70.0% (95% CI, 58.0–81.0%; p = 0.000)) (Supplementary Fig. 59), LGBTQI+ community (72.0% [95% CI, 62.0–81.0%; p = 0.000] to 76.0% (95% CI, 67.0–84.0%; p = 0.000)) (Supplementary Fig. 60), and the general public (48.0% (95% CI, 38.0–57.0%; p = 0.000) to 53.0% (95% CI, 42.0–64.0%)) (Supplementary Fig. 61).

For the intention to vaccinate outcome, our leave-one-out sensitivity analysis showed that no single study had an overriding influence on the overall pooled prevalence of intention to vaccinate, with the pooled estimates varying between 60.0% (95% CI, 54.0–66.0%; p = 0.000) and 62.0% (95% CI: 55.0–68.0%; p = 0.000) (Supplementary Fig. 62). Similarly, we found no evidence of an overriding influence of a single study on the pooled intention rates in our leave-one-out sensitivity analysis for PLHIV (72.0% (95% CI, 59.0–84.0%; p = 0.000) to 78.0% (95% CI, 66.0–88.0%; p = 0.000)) (Supplementary Fig. 63), LGBTQI+ community (75.0% (95% CI, 68.0–82.0%; p = 0.000) to 79.0% (95% CI, 72.0–84.0%; p = 0.000)) (Supplementary Fig. 64), healthcare workers (48.0% (95% CI, 38.0–58.0%; p = 0.000) to 55.0% (95% CI, 45.0–64.0%; p = 0.000)) (Supplementary Fig. 65), and the general public (48.0% (95% CI, 39.0–58.0%; p = 0.000) to 55.0% (95% CI, 43.0–66.0%; p = 0.000)) (Supplementary Fig. 66). However, the leave-one-out analysis found evidence of an overriding influence of the multiregional study by Abd Elhafeez et al.[99] and the Malasian study by Lin et al.[90] (50.0% (95% CI, 118.0–83.0%; p = 0.000) vs. 72.0% (95% CI, 64.0–80.0%; p = 0.000)) on the pooled rate of intention to vaccinate (59.4; 95% CI, 41.8–75.9%) among university students (Supplementary Fig. 67).

The leave-one-out sensitivity analysis for our population-wide mpox vaccine uptake outcome revealed that estimated pooled prevalence of uptake was not influenced by omission of any of the included studies, with the pooled estimates varying between 28.0% (95% CI, 17.0–40.0%; p = 0.000) and 34.0% (95% CI: 22.0–46.0%; p = 0.000) (Supplementary Fig. 68). Similarly, we found no evidence of an overriding influence of any study on the pooled mpox vaccine uptake rates in our leave-one-out sensitivity analysis among PLHIV (34.0% (95% CI, 25.0–43.0%; p = 0.000) to 39.0% (95% CI, 29.0–49.0%; p = 0.000)) (Supplementary Fig. 69), and the LGBTQI+ community (35.0% (95% CI, 28.0–42.0%; p = 0.000) to 42.0% (95% CI, 30.0–54.0%; p = 0.000)) (Supplementary Fig. 70). However, the leave-one-out sensitivity analysis revealed evidence of an overriding influence of the study by Ewijk et al.[44] and the study by Gallè et al.[48] (11.0% (95% CI, 1.0–3.0%; p = 0.000) vs. 26.0% (95% CI, 4.0–59.0%) on the pooled prevalence of uptake (20.2%; 95% CI, 6.9–38.3%) among the general public (Supplementary Fig. 71). Similarly, we found evidence of an overriding influence of the study by Palich et al.[56] and the study by Gallè et al.[48] (29.0% (95% CI, 15.0–46.0%; p = 0.000) vs. 42.0% (95% CI, 24.0–62.0%; p = 0.000)) on the pooled prevalence of vaccine uptake (36.1%; 95% CI, 19.9–54.1) among the accepting group (Supplementary Fig. 72).

## Factors associated with mpox vaccine acceptance and uptake

The most commonly reported correlates of the intention to accept the mpox vaccine and uptake include:

**Age.** Older age has been reported to be associated with higher intention to accept[51,89,97], lower intention to accept[65,93,94,98], and higher hesitancy[95]. Younger age was associated with higher uptake[47,53,96]

**Sex.** Being male is associated with a higher intention to accept[58,60,97], and returning for a second dose[43].

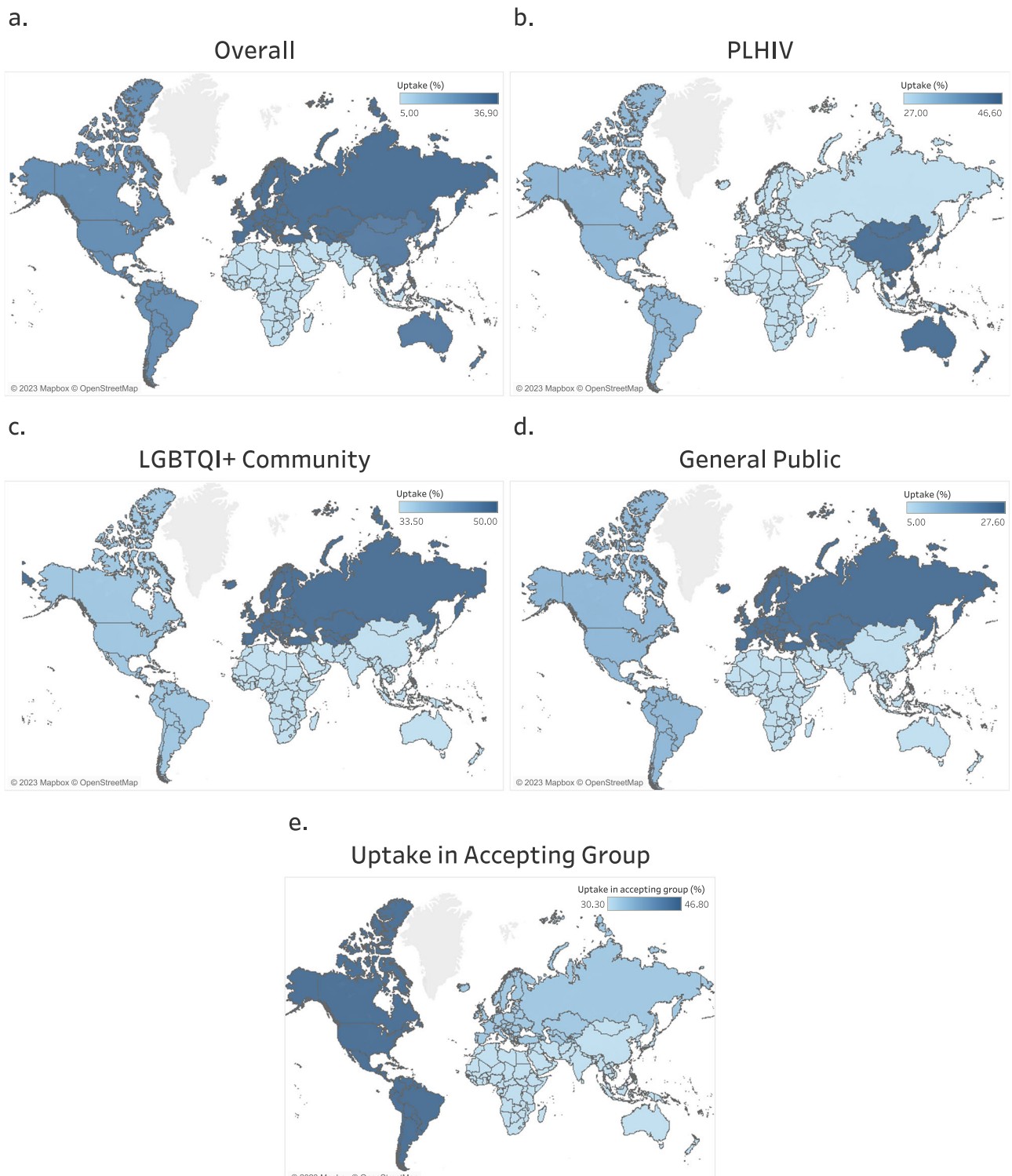

**Fig. 4 | Global map showing the regional pooled prevalence of mpox vaccine uptake across five population groups (overall, PLHIV, LGBTQI+ community, healthcare workers, and the general public).** **a** Population-wide (overall) rates of uptake of the mpox vaccine across all six WHO global regions. **b** Rates of uptake of the mpox vaccine among the accepting group according to WHO global regions (using data available for only three regions: AMR, EUR, and WPR). **c** Rates of uptake of the mpox vaccine among PLHIV according to WHO global regions (using data available for only three regions: AMR, EUR, and WPR). **d** Rates of uptake of the mpox vaccine among the LGBTQI+ community according to WHO global regions (using data available for only three regions: AMR, EUR, and WPR). **e** Rates of uptake of the mpox vaccine among the general public according to WHO global regions (using data available for only three regions: AFR, AMR, and EUR). Darker areas indicate higher rates, and lighter areas indicate lower rates. Maps adapted from OpenStreetMap under a Creative Commons licence CC BY-SA 2.0.

**Level of education**. Attaining a university-level degree has been reported to be associated with both lower[65,87,93,95] and a higher[92,95] likelihood of intention to accept. Similarly, having a below university-level degree has been reported to be associated with a lower likelihood of intention to accept among Chinese MSM[91] and a higher likelihood of intention to accept according to a study among Chinese Healthcare workers[92]. Higher education level has also being reported to be associated with uptake of more than one dose[45].

**Income**. Having a higher income has been reported to be associated with both higher[80,97] and lower[93] likelihood of intention to accept. Low income has been reported to be associated with lower uptake[46].

**Mpox-related concern**. Concern about the mpox[51,65,73–75,77,78,91,93,97,98], including being more worried about the mpox than COVID-19[65,75,93,98], has been reported to be associated with a higher likelihood of intention to accept, while the belief that mpox is being overemphasized is associated with a lower likelihood of intention to accept[51,60].

**Perceived mpox susceptibility**. The perception of being highly susceptible or at risk of mpox has been associated with a higher likelihood of intention to accept the mpox vaccine[45,47,50,51,60,61,74,87,89,91,98].

**Mpox-related knowledge**. Having an above-average level of knowledge about mpox is associated with a higher likelihood of intention to accept[60,75,87,88,92,93,98], and lower hesitancy[95].

**Mpox-related information**. Receiving information about mpox from health authorities is associated with a higher likelihood of intention to accept vaccination[65,75,93,97].

**Mpox vaccine trust**. Trusting the mpox vaccine to be safe is associated with higher acceptance[51,60,63,72,91,94].

**Vaccination history**. Having a good vaccination history has been reported to be associated with higher intention to accept[72] and actual uptake[46]. Previous vaccination against COVID-19[57,60,73,86] and seasonal influenza[82] were associated with a higher likelihood of intention to accept while the refusal of COVID-19 vaccination was associated with lower acceptance[58].

**Mpox vaccine mandate**. Holding the opinion that mpox vaccination be made compulsory for high-risk groups is associated with high intention to accept[78,94].

**Sexual behavior**. Having multiple sexual partners is associated with higher acceptance[50,77,78,85,91], higher uptake[49], and lower hesitancy[79], while being bisexual is associated with lower acceptance[50,80]. On the other hand, engaging in chemsex[74] and using condoms[87] were all associated with a higher likelihood of intention to accept. Being MSM was associated with a higher likelihood of intention to accept[80,86] and a higher uptake[49].

**HIV PrEP**. Being on HIV PrEP is associated with a higher intention[51,74] to accept as well as actual uptake[41,49,53] of the mpox vaccine.

**STI history**. A recent diagnosis of STI within the previous two years has been reported to be associated with higher intention to accept the mpox vaccine[74], and the actual uptake[49,52,53].

**HIV co-infection**. Being HIV-infected has being reported to be associated with higher intention to accept among MSM residing in the EUR[74], while a study among men attending a clinic in Israel[41,46] reported HIV infection to be associated with a lower uptake.

**Comorbidity**. Having a chronic disease has been reported to be associated with a higher likelihood of intention to accept[89,98], but with a lower likelihood of uptake[41] of the mpox vaccine.

## Discussion

This systematic review and meta-analysis evaluated the prevalence of mpox vaccine acceptance and uptake globally, regionally, and across key population subgroups. We also identified the factors associated with vaccine acceptance/uptake. Our findings revealed a suboptimal pooled overall global rate of the mpox vaccine acceptance rate (59.7%). This study also demonstrated substantial global and regional variations in the rates of mpox vaccine acceptance and uptake, overall and across key population groups (PLHIV, LGBTQI+ community, and healthcare workers), with the highest acceptance rate (73.6%) observed among the LGBTQI+ community. Our study also identified several modifiable behavioral factors associated with a higher likelihood of mpox vaccine acceptance, including being concerned about getting infected, the perception of being highly at risk, and knowledge about mpox, as well as receiving information about the disease from healthcare authorities

This systematic review and meta-analysis are, to the best of our knowledge, the largest study reporting the global prevalence of acceptance and uptake of the mpox vaccine, with representation from all WHO regions and almost all countries reporting a confirmed case of the disease. Also, to the best of our knowledge, this is the first meta-analysis to report the global prevalence of mpox vaccine uptake. The pooled overall global rate of the mpox vaccine acceptance rate from all the included studies was 59.7%, The overall pooled global rate of intention to vaccinate against mpox in this study (60.9%) falls within the range of 56.0% (11 studies)[19] and 61.05% (29 studies)[21] reported by previous meta-analyses. The overall global pooled uptake rate was 30.9%, with the LGBTQI+ community having a substantially higher uptake rate (39.8%) than the general public (20.2%). Among PLHIV, the pooled global acceptance and uptake rates were 66.4% (with substantial variations across the WHO regions) and 35.7%.

The findings of our meta-analysis revealed a remarkable gap between the overall global rate of intention to vaccinate against mpox (60.9%) and the overall global mpox vaccine uptake rate (30.9%). Moreover, further analyses of our data showed that only 36.6% of those who indicated their intention to receive the mpox vaccine actually received the vaccine. These findings indicate the existence of a major lag in vaccine uptake among those intending to vaccinate. Several factors, including variations in the timing of the studies and the availability of the vaccine across individual study settings, may have contributed to these observed gaps. Nonetheless, aggressive vaccination policies and strategies may be needed to bridge these gaps and ultimately meet the demands of the unvaccinated people who intend to vaccinate. The need for these policies is particularly crucial because unvaccinated individuals who intend to vaccinate remain at high risk of switching back to being uncertain or refusing to vaccinate[100]. Also, it is worth noting that the observed global overall prevalence of intention to vaccinate against mpox (60.9%) is substantially lower than the rate (95.3%)[101] reported for the malaria vaccine, although it is relatively similar to the rates of intention to vaccinate against COVID-19 widely reported in previous studies (range: 60% − 65%)[102–105]. Similarly, the observed rates of the overall mpox vaccine uptake (30.9%) were relatively lower than the overall rates of the COVID-19 vaccine uptake (42.3%) reported in a previous large–scale meta-analysis[25]. The variations in rates of uptake and intention to vaccinate across these diseases may be explained by the differences in the relative prevalence, severity, and case fatality associated with the diseases.

Furthermore, our meta-analysis demonstrated that the global prevalence of mpox vaccine acceptance among PLHIV (66.4%) is comparable to that of the COVID-19 vaccine (67.0%)[106]. We also found that the global intention to vaccinate against mpox among PLHIV (75.0%) and the actual vaccine uptake (35.7%) were substantially higher than the observed pooled global overall rates of the mpox vaccine intention (60.9%) and uptake (30.9%). Furthermore, given that the prevalence of HIV coinfection among

patients with mpox has been reported to be as high as 40%[14,15,17,18], a high rate of intention to be vaccinated against mpox among PLHIV may substantially lower the overall prevalence of mpox if appropriate measures are put in place to improve vaccine uptake among PLHIV, particularly those who are already intending to be vaccinated.

The results of our meta-analysis also revealed considerable variations in the overall and population-specific rates of acceptance, intention, and uptake of the mpox vaccine across WHO regions. The overall acceptance rate for the WPR was 72.2%, the SEAR had 67.3%, EUR had 63.8%, the EMR had 52.0%, AMR had 48.9%, and AFR had 41.9%, while the rate of uptake was 46.8% among the accepting group for the AMR, 33.7% among those in EUR, and 30.3% among those in the WPR. The overall intention to vaccinate and the actual vaccine uptake were 73.5% and 33.5% for the WPR, 59,3% and 36.9% for EUR, 44.8% and 28.3% for AMR, and 41.9% and 5.0% in AFR. These findings have important implications. First, compared to the WPR and EUR, the WHO AFR had the lowest overall rate of intention and uptake despite being home to mpox-endemic countries, like Nigeria, which has been the source of most mpox outbreaks, including the 2022 outbreak that started in the UK[107]. Therefore, strong public health policies specific to mpox awareness and prevention are needed particularly in the WHO African region to prevent future outbreaks. Second, a relatively weaker health system in the WHO AFR may explain the endemicity of the disease and outbreaks in countries within the region. Therefore, as part of strengthening the global health system, building capacity for disease surveillance, emergency preparedness and response in the WHO AFR has been suggested as a potent means of substantially rolling back the spectrum of the mpox endemicity in the region[108,109]. Furthermore, vaccinating animals in settings with the confirmed animal-to-human transmission may be employed to successfully eradicate the disease[108].

The higher rates of acceptance and uptake observed among the LGBTQI+ community (73.6% and 39.8%, respectively – relative to the population-wide average of 49% acceptance and 11% uptake rate – may indicate the group's higher risk perception and better awareness compared to the general public. Among healthcare workers, who are also at a high risk of contracting mpox, the prevalence of intention to vaccinate against mpox (51.9%) is comparable to the acceptance rate of the COVID-19 vaccine (55.9%-65.7)[104,105,110]. However, the prevalence of intention to vaccinate among the general public, considered to be at a lower risk of mpox is substantially lower than the rate reported for the COVID-19 vaccine (52.3% for mpox vs. 61.0%-81.65% for COVID-19)[104,105,110]. These findings further illustrate the potential role of risk perception on vaccine acceptance. Importantly, despite the WHO's recommendation of vaccination against mpox by the high-risk LGBTQI+ community[111], only 36.1% of those intending to receive the vaccine had taken one or more doses of the vaccine against mpox either as PPV or PEPV. Although the proportion of uptake among those intending to be vaccinated is substantially higher for the LGBTQI+ community (39.8%) compared to the population-wide average of 30.9%, the high burden of mpox among the LGBTQI+ implies that vaccine uptake among this vulnerable group is still suboptimal. Therefore, further research is needed to develop strategies to improve vaccine uptake to meet the vaccination demands of unvaccinated LGBTQI+ community members who intend to get vaccinated.

Furthermore, our narrative synthesis highlighted important correlates of the acceptance of the mpox vaccine. Age, sex, level of education, and level of income are among the most reported sociodemographic determinants of mpox vaccine acceptance. Therefore, public health intervention programs aimed at enhancing positive community attitudes toward the mpox vaccination program need to consider these sociodemographic characteristics in order to maximize acceptance and uptake of the vaccine. Our review has shown that having a few sexual partners and being bisexual are associated with lower vaccine acceptance rate; therefore, it is vital that intervention programs take into account the sexual behaviors of the target population. Furthermore, our results showed that MSM who are not on HIV PrEP have lower acceptance of the mpox vaccination. This finding indicates the need for deliberate health education and awareness efforts aimed at addressing vaccination hesitancy in this high-risk group. Moreover, MSM who are not on HIV PrEP may be targeted for HIV testing during mpox vaccination and appropriately counseled for commencement of PrEP if found not positive for HIV, as recommended[112].

Finally, this review also found some important modifiable behavioral factors associated with mpox vaccine acceptance, including concerns about the disease, the perception of being highly at risk, and knowledge about mpox, as well as the source of information about the disease. These findings indicate the critical need for behavioral interventions to increase knowledge and clear misperceptions related to mpox in the community, especially among high-risk groups such as the LGBTQI+ community. Accordingly, these interventions need to incorporate measures that favor public receipt of mpox-related information from reliable sources like health institutions and/or professionals. This consideration is particularly important given that the current 2022 mpox outbreak occurred during the COVID-19 pandemic era, a period that has been characterized by unprecedented levels of vaccine hesitancy, mostly fueled by a phenomenon termed "infodemic", defined by the WHO as "too much information, including false or misleading information in digital and physical environments during a disease outbreak"[113] As evidence has strongly linked infodemic to vaccine hesitancy[100,114,115], efforts to maximize the mpox vaccine acceptance also need to include measures to combat misinformation-induced hesitancy, especially the misinformation spread via online media. Active engagement of public health professionals and institutions in online media campaigns has been recommended as one of the potent ways of addressing this growing problem[116]. Also, since an individual's level of trust in the mpox vaccine and previous vaccination against COVID-19 and influenza history are also strong correlates of mpox vaccine acceptance, efforts are needed to accelerate public trust in vaccines, and emphasis should be given to those individuals with poor vaccination history. Also, because trust in vaccines is a highly delicate topic, caution must be undertaken before enacting mandatory vaccine policies, as these policies may fester hesitancy, further spawn the growing anti-vaccine activism, and potentially expel some individuals who were previously intent on getting vaccinated. Of note, we did not find a single study conducted in the WHO AFR that reported the specific prevalence of vaccine intention, acceptance, or uptake among the LGBTQI+, even though this population has been numerously identified as a high-risk group and mpox is known to be endemic in many countries in the region. Therefore, future studies from this region should focus on evaluating the prevalence of and factors associated with mpox vaccine intention, uptake, and acceptance among the LGBTQI+ communities.

Among the key strengths of this review is a literature search involving multiple databases, which provided a high number of included studies (sixty-one) having a cumulative sample size of 263,857 from 87 countries across all six WHO regions. Second, our critical appraisal showed that the majority of the included studies have high methodological rigor. Third, this study is the first meta-analysis to evaluate the gap between acceptance and uptake, by evaluating the rate of vaccine uptake among those who indicated vaccine acceptance. The limitations of this review are mostly related to the included studies, including the utilization of online surveys by a vast majority of the included studies, which may introduce participant recruitment bias, and exclude people with no/or limited access to the internet. Second, there is high degree of statistical heterogeneity in the overall analysis, which remained present when we disaggregated the data for each of the population groups and performed a subgroup analysis by region. However, we have performed and reported the results of our leave-one-out sensitivity analysis for each of the three major outcomes across all population groups. Third, nearly all of the studies employed a non-probability sampling technique, which may be associated with selection bias. Fourth, due to limited availability of funds for cross-language translations, only studies published in the English language were considered, thereby potentially limiting the generalizability of our pooled estimates. Fifth, some population regions, like the SEAR have only three studies, further limiting the generalizability of the sub-group analyses by WHO regions.

## Conclusion

This review demonstrated the existence of substantial regional variations in the rates of mpox vaccine acceptance and uptake, as well as the presence of a wide gap between the rate of vaccine acceptance and vaccine uptake. Among the LGBTQI+ community, a group designated as a high-risk group for mpox, only about one-third of those who indicated vaccine acceptance actually received at least a single dose, and an even wider acceptance-uptake gap was reported for the general population. Targeted intervention programs to maximize mpox vaccine uptake, which account for the socio-demographic and other behavioral predictors of low mpox vaccine acceptance, particularly among high-risk groups are needed to reduce the overall global burden of mpox.

## Data availability

All the source data supporting the results presented in this work is accessible at https://doi.org/10.17605/OSF.IO/FS5QH.

## Code availability

All the code for the data analyzed in this work is openly available at Open Science Framework (via https://doi.org/10.17605/OSF.IO/FS5QH)[117].

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

## Acknowledgements
The authors solely supported this work. No external funding was received for the work.

## Author contribution
S.K.S. conceived the study. S.K.S., M.S.M., B.T.M., F.I.T., A.K.S., and A.T.B. all contributed to the study methodology. S.K.S., M.S.M., and A.T.B. performed all statistical analysis. S.K.S., M.S.M., and A.T.B. were responsible for data curation. S.K.S. prepared the original manuscript draft; A.T.B., M.S.M., and S.K.S. led the critical review and editing of subsequent manuscript drafts along with contributions from F.I.T., A.K.S., and B.T.M.; A.K.S. led the study supervision; All authors contributed resources. All authors contributed to study validation. A.T.B. and S.K.S. were responsible for study visualization. S.K.S. was responsible for the project administration.

## Competing interest
The authors declare no competing interests.
