## [Peer Review File · Communications Medicine]

Reviewers' comments:

Reviewer #1 (Remarks to the Author):

Dear Editor,

I would like to express my gratitude for the opportunity to review this paper. I would also like to acknowledge the authors for their valuable contributions. Despite the existence of two similar studies, the authors undertook this review to include additional studies and address further objectives related to Mpox vaccination.

However, I have several comments to suggest improvements to the quality of this work:

Major comments:

A significant proportion of the pooled analyses showed substantial heterogeneity, which hindered the interpretation of the findings. I recommend that the authors investigate the sources of this heterogeneity by either rechecking the data extraction process or considering the exclusion of low-quality studies. It would be advisable to avoid pooling studies with different objectives and instead conduct subgroup analyses based on the tools used. Additionally, employing advanced analyses such as Gosh sensitivity analysis and metaregression may provide valuable insights. I suggest referring to the following useful paper for guidance: [<https://www.nature.com/articles/s41598-021-04345-x>].

The excessive use of abbreviations, especially in the abstract, detracts from readability. It is recommended to limit the use of abbreviations or provide their full forms upon the first mention.

The authors did not mention how disagreements among the reviewers were resolved or the value of the kappa test of agreement between reviewers. Including this information would enhance transparency and indicate the reliability of the review process.

Despite reporting publication bias in only one outcome, the authors did not search the grey literature or include non-published data. It is crucial to consider these additional sources of information to improve the comprehensiveness of the review.

The authors did not specify whether they conducted a manual search and citation tracking. Clarifying whether these methods were employed would provide a more comprehensive understanding of the search strategy.

In lines 169-171, combining the acceptance group (intention + uptake) may be a source of heterogeneity. It is advisable to treat these outcomes separately to avoid potential confounding factors.

In the discussion section, please provide a summary of the studies included to provide a concise overview for readers.

In addition to the major comments, I have a few minor suggestions:

Please update the figures related to Mpox in the abstract to ensure accuracy.

It would be beneficial to mention the efficacy and generations of vaccines primarily used for smallpox and Mpox.

Although hesitancy is mentioned in the introduction, it is not defined. It would be helpful to provide a clear definition for readers' understanding.

If applicable, I suggest updating the search as more than three months have passed since the initial search was conducted.

In line 157, please clarify why two references were used.

In line 159, please correct the reference by removing the "p" at the end.

please revise the comment on the extraction table; During the extraction process, Algeria was included in three studies, not two. Although you mentioned two of them, reference 46 also includes Algeria.

Please report the heterogeneity alongside the pooled effect size and confidence interval to provide a comprehensive analysis.

Improve the resolution of the figures to ensure clear visualization.

I believe addressing these comments and suggestions will significantly improve the manuscript.

Thank you once again for the opportunity to review this paper.

Reviewer #2 (Remarks to the Author):

The article discussing the global acceptance and uptake of the mpox vaccine is undeniably intriguing, offering valuable insights into a crucial aspect of public health. However, I must express my concerns, particularly regarding the statistical analysis/methodology employed in the study. Before considering acceptance for publication, the following concerns must be addressed:

1. In line 72 of the manuscript the authors make a reference to previous studies of vaccination intention in mpox, however, they cite a study carried out for COVID-19. Please make the corresponding corrections

2. In line 161-162 the authors indicate that they used a Freeman-tukey double arcsine transformation to include studies with proportions close to or equal to 1, however, in these cases the correct thing to do is to use a continuity correction and not a variance stabilizer.

3. In lines 174 to 176, the authors indicate having used the Egger's test and Begg's test in search of evidence of publication bias. However, this is not a practice recommended by the scientific literature regarding meta-analyses of proportions.

(<https://bmcmmedresmethodol.biomedcentral.com/articles/10.1186/s12874-021-01381-z>)

(<https://pubmed.ncbi.nlm.nih.gov/24794697/>)

4. In Table 1, the article by Sagy et al. [Citation number 60]), is a retrospective cohort, however, when reviewing the section of the manuscript "Critical appraisal (quality assessment) of included studies" and observing the quality assessment scoring table, the authors only used the Newcastle-Ottawa Scale (NOS) for Cross-sectional studies. Please, it is necessary to correct this and evaluate the article in question using a specific scale for cohort studies.

5. In the discussion, the authors should compare the results of their study with previous systematic reviews on vaccination acceptance for mpox.

Reviewer #3 (Remarks to the Author):

I would thank the authors for this interesting work. However, after reviewing, I found some concern that bothered me:

1. The first concern is related to the fact that the authors included some studies conducted in 2020 which could affect the results. In fact, the COVID-19 pandemic has given a new vision of this phenomenon. Also, including 2 studies among 42 could affect the results.

2. The second concern is related to the manner of writing. In fact, the authors used nearly the same expression (among....) in almost all the results.

3. Some spelling errors are also observed (exp: line 16- 17: EUR 17.9%; AMR 13.3%; and AFR 5.0%." add the brackets;
line 17-18: Acceptance and uptake were 72.0% and 30.1% among the LGBTQI+ community, and healthcare workers (59.8%)" correct please

Line 159: delete p3

Line 224-239: the term "by both Begg's test and Egger's test" is repeated multiple times. try to reformulate. Also "p" value should be in lower case (in all the manuscript).

The same remark for lines 243-252 and 265-275 (then is repeated multiple times). Try to reformulate please.

Also, you do not need to repeat "metaanalysis" in each tile, you can use just: "prevalence...".

At last, I suggest to reformulate the title

Good luck

REVIEWER 1 COMMENTS

I would like to express my gratitude for the opportunity to review this paper. I would also like to acknowledge the authors for their valuable contributions. Despite the existence of two similar studies, the authors undertook this review to include additional studies and address further objectives related to Mpox vaccination.

However, I have several comments to suggest improvements to the quality of this work:

Major concerns

Comment (1)

A significant proportion of the pooled analyses showed substantial heterogeneity, which hindered the interpretation of the findings. I recommend that the authors investigate the sources of this heterogeneity by either rechecking the data extraction process or considering the exclusion of low-quality studies. It would be advisable to avoid pooling studies with different objectives and instead conduct subgroup analyses based on the tools used. Additionally, employing advanced analyses such as Gosh sensitivity analysis and metaregression may provide valuable insights. I suggest referring to the following useful paper for guidance: [<https://www.nature.com/articles/s41598-021-04345-x>].

Response

We thank the reviewer for this feedback. We want to note that “ I^2 ” is usually known to be high in meta-analyses of proportions (such as in this study where the objective is not to compare two groups but to derive a single summary estimate of the individual proportions reported in the included studies). Barker et al. (2021) -<https://doi.org/10.1186/s12874-021-01381-z> – explained as follows:

“Although I^2 was developed in the context of comparative data, it is commonly applied to estimate heterogeneity for proportional meta-analysis. In this type of analysis, I^2 is usually high. This can be due to the nature of proportional data, where little variance is observed even in studies with small sample sizes. Moreover, true heterogeneity is expected in prevalence and incidence estimates due to differences in the time and place where included studies were conducted. Therefore, high I^2 in the context of proportional meta-analysis does not necessarily mean that data is inconsistent.”

Nonetheless, we explored the included studies, per the reviewer’s recommendation, to ascertain if the inclusion of low-quality studies may have led to the observed heterogeneity. However, none of the included studies we scored is of a low quality that warrants its exclusion from the analysis. Moreover, just as found in the paper cited by the reviewer, the exclusion of low-quality studies does not necessarily eliminate heterogeneity. Also, we’ve rechecked our data and ascertained that all included studies conform with our inclusion criteria and have similar objectives with respect to the corresponding study outcomes (intention, uptake, and acceptance [intention + uptake]). We further confirmed that the extracted values are accurate. Moreover, we treated each of these outcomes independently and conducted a meta-analysis for each.

On the issue of performing “subgroup analysis based on the tools used,” we wish to note that all our study outcomes were defined according to the previous studies on vaccine acceptance, intention, and uptake. Although some of the included studies used different scales to define the study outcomes, we only used the actual frequencies/proportions of intention/uptake/acceptance from these studies in accordance with our

pre-specified outcome definitions. Thus, we defined the prevalence of intention to be vaccinated as the proportion of those who responded “yes” (for a binary question, yes or no) or “yes definitely/yes likely” (for a four- or five-point Likert scale question) to a question asking participants about their willingness to receive the vaccine. We defined uptake prevalence as the proportion of those who received at least one dose of any of the recommended mpox vaccines among the study population, while the prevalence of acceptance was defined as the proportion of those who intend to receive the vaccine and those who have taken at least a single dose among the study population.

Also, per the reviewer’s feedback, we have expanded the scope of our subgroup analyses for each outcome and evaluated the overall and regional prevalence further stratified by study subpopulations (LGBTQI+ community, health workers, PLHIV, and the general public). This new analysis more deeply explores the potential sources of heterogeneity between the included studies.

We also agree with the reviewer’s recommendation for a sensitivity analysis. We have performed a series of leave-one-out meta-analyses for each outcome (overall and stratified by population subgroups, where possible) to assess if the omission of any of the included studies has an overriding influence on the overall estimate. The results of these analyses are presented in the results section of the manuscript.

Comment (2)

The excessive use of abbreviations, especially in the abstract, detracts from readability. It is recommended to limit the use of abbreviations or provide their full forms upon the first mention.

Response

We thank the Reviewer for this feedback. We've re-read the whole manuscript to ascertain that all abbreviations are written in their full forms on the first mention. In addition, we have also created a dictionary of all abbreviations used and added it at the end of the manuscript.

Comment (3)

The authors did not mention how disagreements among the reviewers were resolved or the value of the kappa test of agreement between reviewers. Including this information would enhance transparency and indicate the reliability of the review process.

Response

We thank the Reviewer for this important feedback. We've provided information on how disagreements were resolved on page- 9, lines 204-207. Where disputes occurred between the two junior authors (SKS and MSM) critically appraising the studies, we relied on consensus between our two senior authors for a final consensus score (as done in the paper cited by the reviewer). Below is an excerpt from the manuscript detailing how we addressed disagreement:

“The scores of the two investigators were compared and reviewed by two senior authors (FIT and ATB), and where disputes occurred, a final consensus score was decided by the senior authors through revision and discussion of the articles together.”

Comment (4)

Despite reporting publication bias in only one outcome, the authors did not search the grey literature or include non-published data. It is crucial to consider these additional sources of information to improve the comprehensiveness of the review.

Response

We thank the Reviewer for this suggestion. We've repeated our literature search in the select databases and added three more databases (the Regional Office for Africa Library, the African Index Medicus, and the WHO Institutional Repository for Information Sharing) and also employed forward and backward citation tracking to include studies published by 30th October 2023. However, we did not include preprints, grey literature, and other non-peer-reviewed studies, as suggested by the reviewer, because the Editor required that included studies should be limited to peer-reviewed articles only. We've also revised our inclusion criteria to "include only peer-reviewed publications".

We also wish to highlight that the seven studies that were available as preprints in our initial submission have now all been published as peer-reviewed articles.

Below is the list of the seven studies that have now been published as peer-reviewed articles.

1. Al-Mustapha AI, Ogundijo OA, Sikiru NA, et al. A cross-sectional survey of public knowledge of the monkeypox disease in Nigeria. *BMC Public Health*. 2023;23(1):591. doi:10.1186/s12889-023-15398-0
2. Swed S, Alibrahim H, Bohsas H, et al. A multinational cross-sectional study on the awareness and concerns of healthcare providers toward monkeypox and the promotion of the monkeypox vaccination. *Front Public Health*. 2023;11. Accessed November 3, 2023.<https://www.frontiersin.org/articles/10.3389/fpubh.2023.1153136>.
3. MacGibbon J, Cornelisse VJ, Smith AKJ, et al. Mpox (monkeypox) knowledge, concern, willingness to change behaviour, and seek vaccination: results of a national cross-sectional survey. *Sex Health*. 2023;20(5):403-410. doi:10.1071/SH23047.
4. Swed S, Bohsas H, Alibrahim H, et al. Monkeypox Post-COVID-19: Knowledge, Worrying, and Vaccine Adoption in the Arabic General Population. *Vaccines*. 2023;11(4):759. doi:10.3390/vaccines11040759 47.
5. Smith LE, Potts HW, Brainard J, et al. Did mpox knowledge, attitudes and beliefs affect intended behaviour in the general population and men who are gay, bisexual and who have sex with men?

An online cross-sectional survey in the UK. *BMJ Open*. 2023;13(10):e070882. doi:10.1136/bmjopen-2022-070882

6. Peptan C, Băleanu VD, Mărcău FC. Study on the Vaccination of the Population of Romania against Monkeypox in Terms of Medical Security. *Vaccines*. 2022;10(11):1834. doi:10.3390/vaccines10111834

7. Alarifi AM, Alshahrani NZ, Sah R. Are Saudi Healthcare Workers Willing to Receive the Monkeypox Virus Vaccine? Evidence from a Descriptive-Baseline Survey. *Trop Med Infect Dis*. 2023;8(8):396. doi:10.3390/tropicalmed8080396

Furthermore, we wish to note that our updated literature search resulted in the addition of 21 new peer-reviewed studies to the data. Please see Table 1 in supplementary information for details of these studies. Below is a list of the references for the studies.

1. Abd ElHafeez S, Gebreal A, Khalil MA, et al. Assessing disparities in medical students' knowledge and attitude about monkeypox: a cross-sectional study of 27 countries across three continents. *Front Public Health*. 2023;11:1192542. doi:10.3389/fpubh.2023.1192542
2. Islam MR, Haque MA, Ahamed B, et al. Assessment of vaccine perception and vaccination intention of Mpox infection among the adult males in Bangladesh: A cross-sectional study findings. *PLOS ONE*. 2023;18(6):e0286322. doi:10.1371/journal.pone.0286322
3. Riad A, Rybakova N, Dubatouka N, et al. Belarusian Healthcare Professionals' Views on Monkeypox and Vaccine Hesitancy. *Vaccines*. 2023;11(8):1368. doi:10.3390/vaccines11081368
4. Salih T. Demographic Disparities in Mpox Vaccination Series Completion, by Route of Vaccine Administration — California, August 9, 2022–March 31, 2023. *MMWR Morb Mortal Wkly Rep*. 2023;72. doi:10.15585/mmwr.mm7230a4
5. Torres TS, Silva MST, Coutinho C, et al. Evaluation of Mpox Knowledge, Stigma, and Willingness to Vaccinate for Mpox: Cross-Sectional Web-Based Survey Among Sexual and Gender Minorities. *JMIR Public Health Surveill*. 2023;9(1):e46489. doi:10.2196/46489
6. JAMALEDDINE Y, EL EZZ AA, MAHMOUD M, et al. Knowledge and attitude towards monkeypox among the Lebanese population and their attitude towards vaccination. *J Prev Med Hyg*. 2023;64(1):E13-E26. doi:10.15167/2421-4248/jpmh2023.64.1.2903 73.
7. van Ewijk CE, Smit C, Bavalia R, et al. Acceptance and timeliness of post-exposure vaccination against mpox in high-risk contacts, Amsterdam, the Netherlands, May–July 2022. *Vaccine*. 2023;41(47):6952-6959. doi:10.1016/j.vaccine.2023.10.013

8. Castel AD, Andersen E, Monroe A, et al. Mpox Awareness, Risk Reduction, and Vaccine Acceptance among Pwh in Washington, Dc. *Top Antivir Med.* 2023:401-402.
9. Svartstein ASW, Knudsen AD, Heidari SL, et al. Mpox Incidence and Vaccine Uptake in Men Who Have Sex with Men and Are Living with HIV in Denmark. *Vaccines.* 2023;11(7):1167. doi:10.3390/vaccines11071167
10. Chow EPF, Samra RS, Bradshaw CS, et al. Mpox knowledge, vaccination and intention to reduce sexual risk practices among men who have sex with men and transgender people in response to the 2022 mpox outbreak: a cross-sectional study in Victoria, Australia. *Sex Health.* Published online July 10, 2023. doi:10.1071/SH23075 77.
11. Zheng M, Du M, Yang G, et al. Mpox Vaccination Hesitancy and Its Associated Factors among Men Who Have Sex with Men in China: A National Observational Study. *Vaccines.* 2023;11(9):1432. doi:10.3390/vaccines11091432.
12. Filardo TD, Prasad N, Waddell CJ, et al. Mpox vaccine acceptability among people experiencing homelessness in San Francisco — October–November 2022. *Vaccine.* 2023;41(39):5673-5677. doi:10.1016/j.vaccine.2023.07.068
13. Araoz-Salinas JM, Ortiz-Saavedra B, Ponce-Rosas L, et al. Perceptions and Intention to Get Vaccinated against Mpox among the LGBTIQ+ Community during the 2022 Outbreak: A Cross-Sectional Study in Peru. *Vaccines.* 2023;11(5):1008. doi:10.3390/vaccines11051008
14. Curtis MG, Davoudpour S, Rodriguez-Ortiz AE, et al. Predictors of Mpox vaccine uptake among sexual and gender minority young adults living in Illinois: Unvaccinated vs. double vs. single dose vaccine recipients. *Vaccine.* 2023;41(27):4002-4008. doi:10.1016/j.vaccine.2023.05.043
15. Mahameed H, Al-Mahzoum K, AlRaie LA, et al. Previous Vaccination History and Psychological Factors as Significant Predictors of Willingness to Receive Mpox Vaccination and a Favorable Attitude towards Compulsory Vaccination. *Vaccines.* 2023;11(5):897. doi:10.3390/vaccines11050897
16. Caycho-Rodríguez T, Tomás JM, Vilca LW, et al. Relationship Between Fear of Monkeypox and Intention to be Vaccinated Against Monkeypox in a Peruvian Sample. The Mediating Role of Conspiracy Beliefs About Monkeypox. *Eval Health Prof.* Published online May 29, 2023:01632787231180195. doi:10.1177/01632787231180195
17. Hori D, Kaneda Y, Ozaki A, Tabuchi T. Sexual orientation was associated with intention to be vaccinated with a smallpox vaccine against mpox: A cross-sectional preliminary survey in Japan. *Vaccine.* 2023;41(27):3954-3959. doi:10.1016/j.vaccine.2023.05.050 84.
18. Gilbert M, Ablona A, Chang HJ, et al. Uptake of Mpox vaccination among transgender people and gay, bisexual and other men who have sex with men among sexually-transmitted infection clinic clients in Vancouver, British Columbia. *Vaccine.* 2023;41(15):2485-2494. doi:10.1016/j.vaccine.2023.02.075

19. Zucker R, Wolff-Sagy Y, Ramot N, et al. Examining the Patterns of Mpox Vaccine Uptake in a Vulnerable Population. *Sex Transm Dis.*:10.1097/OLQ.0000000000001839. doi:10.1097/OLQ.0000000000001839
20. Palich R, Jedrzejewski T, Schneider L, et al. High uptake of vaccination against mpox in men who have sex with men (MSM) on HIV pre-exposure prophylaxis (PrEP) in Paris, France. *Sex Transm Infect.* Published online July 28, 2023. doi:10.1136/sextrans-2023-055885
21. Abara WE, Sullivan P, Carpino T, et al. Characteristics of mpox vaccine recipients among a sample of men who have sex with men with presumed exposure to mpox. *Sex Transm Dis.* 2023;50(7):458-461. doi:10.1097/OLQ.0000000000001800 88.

Comment (5)

The authors did not specify whether they conducted a manual search and citation tracking. Clarifying whether these methods were employed would provide a more comprehensive understanding of the search strategy.

Response

We thank the Reviewer for this important comment. We've employed manual search as well as forward and backward citation tracking to retrieve studies and have included this update to our methods section (page 7/ lines 156-157) and the PRISMA flow diagram for study selection.

Comment (6)

In lines 169-171, combining the acceptance group (intention + uptake) may be a source of heterogeneity. It is advisable to treat these outcomes separately to avoid potential confounding factors.

Response

We thank the Reviewer for this important suggestion. We defined acceptance in accordance with previously published studies (1. Wang, Q., Hu, S., Du, F. *et al.* Mapping global acceptance and uptake of COVID-19 vaccination: A systematic review and meta-analysis.

Commun Med 2, 113 (2022). <https://doi.org/10.1038/s43856-022-00177-6>; 2. Sulaiman, S.K., Musa, M.S., Tsiga-Ahmed, F.I. *et al.* A systematic review and meta-analysis of the global prevalence and determinants of COVID-19 vaccine acceptance and uptake in people living with HIV. *Nat Hum Behav* (2023). <https://doi.org/10.1038/s41562-023-01733-3>; 3. Abdelmoneim, S.A.; Sallam, M.; Hafez, D.M.; Elrewany, E.; Mousli, H.M.; Hammad, E.M.; Elkhadry, S.W.; Adam, M.F.; Ghobashy, A.A.; Naguib, M.; et al. COVID-19 Vaccine Booster Dose Acceptance: Systematic Review and Meta-Analysis. *Trop. Med. Infect. Dis.* **2022**, *7*, 298. <https://doi.org/10.3390/tropicalmed7100298>). Moreover, we have performed a separate analysis for intention and uptake. We have more clearly explained this in the methods section (page 10, line 223-231).

Comment (7)

In the discussion section, please provide a summary of the studies included to provide a concise overview for readers.

Response

We thank the Reviewer for this important suggestion. We have added a summary of the included studies at the beginning of the discussion section, as recommended (page 25, lines 589-599).

Minor Comments

Comment (8)

Please update the figures related to Mpox in the abstract to ensure accuracy.

Response

We thank the Editor for this important suggestion. We've updated the figures as suggested. Please see the abstract for confirmation.

Comment (9)

It would be beneficial to mention the efficacy and generations of vaccines primarily used for smallpox and Mpox.

Response

We thank the Reviewer for this suggestion. We've added detailed information regarding the approved vaccines in the second paragraph of our introduction, as recommended. We added a comment on the safety, efficacy, and immunogenicity of the mpox vaccines based on the findings of a most recently published systematic review study. Below is an excerpt from the second paragraph of our introduction section:

Vaccination against mpox may be provided to individuals at high risk of the infection as primary preventive vaccination (PPV) before exposure to the mpox virus, or as post-exposure preventive vaccination (PEPV), for contacts of mpox cases.^{2,5} In addition to the previously employed smallpox vaccine, which is highly effective in protecting against mpox,¹¹ newer vaccines, including the MVA-BN, LC16, and the ACAM2000 have been approved in many countries for the prevention of mpox.^{2,12} A recently published systematic review shows that these vaccines are highly effective, safe, and immunogenic depending on the number of doses administered and that vaccines against smallpox offer cross-protection against mpox.¹³

2. Vaccines and immunization for monkeypox: Interim guidance, 16 November 2022. Accessed February 18, 2023. <https://www.who.int/publications-detail-redirect/WHO-MPX-Immunization>

5. Mpox (Monkeypox). Accessed February 19, 2023. <https://www.ecdc.europa.eu/en/mpox-monkeypox>

11. Monkeypox-WHO Fact Sheet. Accessed February 18, 2023. <https://www.who.int/news-room/fact-sheets/detail/monkeypox>

12. CDC. Mpox and HIV. Centers for Disease Control and Prevention. Published October 31, 2022. Accessed February 18, 2023. <https://www.cdc.gov/poxvirus/monkeypox/prevention/hiv.html>

13. Ghazy RM, Elrewany E, Gebreal A, et al. Systematic Review on the Efficacy, Effectiveness, Safety, and Immunogenicity of Monkeypox Vaccine. *Vaccines*. 2023;11(11):1708. doi:10.3390/vaccines11111708

Comment (10)

Although hesitancy is mentioned in the introduction, it is not defined. It would be helpful to provide a clear definition for readers' understanding.

Response

We thank the Reviewer for this suggestion. We have added a definition of vaccine hesitancy, as provided by the WHO Strategic Advisory Group of Experts on Vaccine hesitancy (page 4, lines 84-86):

However, this outbreak comes at a time when the world is witnessing all-time high levels of vaccine hesitancy,⁶⁻⁹ defined by the WHO as “a delay in the acceptance or refusal of vaccination despite the availability of vaccination services”¹⁰

References:

6. Bergen N, Kirkby K, Fuertes CV, et al. Global state of education-related inequality in COVID-19 vaccine coverage, structural barriers, vaccine hesitancy, and vaccine refusal: findings from the Global COVID-19 Trends and Impact Survey. *Lancet Glob Health*. 2022;0(0). doi:10.1016/S2214-109X(22)00520-4

7. Wiegand M, Eagan RL, Karimov R, Lin L, Larson HJ, Figueiredo A de. Global Declines in Vaccine Confidence from 2015 to 2022: A Large-Scale Retrospective Analysis. Published online May 8, 2023. doi:10.2139/ssrn.4438003

8. Eagan RL, Larson HJ, de Figueiredo A. Recent trends in vaccine coverage and confidence: A cause for concern. *Hum Vaccines Immunother*. 2023;19(2):2237374. doi:10.1080/21645515.2023.2237374

9. de Figueiredo A, Temfack E, Tajudeen R, Larson HJ. Declining trends in vaccine confidence across sub-Saharan Africa: A large-scale cross-sectional modeling study. *Hum Vaccines Immunother.* 2023;19(1):2213117. doi:10.1080/21645515.2023.2213117

10. MacDonald NE. Vaccine hesitancy: Definition, scope and determinants. *Vaccine.* 2015;33(34):4161-4164. doi:10.1016/j.vaccine.2015.04.036

Comment (11)

If applicable, I suggest updating the search as more than three months have passed since the initial search was conducted.

Response

We thank the Reviewer for this important suggestion. Accordingly, we have updated our literature search to August 25th November, 2023.

Comment (12)

In line 157, please clarify why two references were used.

Response

We thank the Reviewer for this observation. We've correctly maintained one of the references:

Higgins, J. P. et al. Cochrane Handbook for Systematic Reviews of Interventions (John Wiley & Sons, 2019).

Comment (13)

In line 159, please correct the reference by removing the "p" at the end.

Response

We thank the Reviewer for this comment. The “p” has been removed from the citation (page 9, line 217).

Comment (14)

please revise the comment on the extraction table; During the extraction process, Algeria was included in three studies, not two. Although you mentioned two of them, reference 46 also includes Algeria.

Response

We thank the Reviewer for this important suggestion. Two of the three studies that included Algeria were multiregional studies, and these two were combined with other multiregional studies (that provided no full disaggregated data for each of the included countries) in our subgroup analyses by region as “multiregional studies”.

Comment (15)

Please report the heterogeneity alongside the pooled effect size and confidence interval to provide a comprehensive analysis.

Response

We thank the Reviewer for this important suggestion. The individual heterogeneity from all meta-analyses has now been reported and put in the forest plots of the meta-analysis findings.

Comment (16)

Improve the resolution of the figures to ensure clear visualization.

Response

We thank the Reviewer for this important suggestion. We have provided all our figures as high-resolution images.

REVIEWER 2 COMMENTS

The article discussing the global acceptance and uptake of the mpox vaccine is undeniably intriguing, offering valuable insights into a crucial aspect of public health. However, I must express my concerns, particularly regarding the statistical analysis/methodology employed in the study. Before considering acceptance for publication, the following concerns must be addressed:

Comment (1)

1. In line 72 of the manuscript the authors make a reference to previous studies of vaccination intention in mpox, however, they cite a study carried out for COVID-19. Please make the corresponding corrections

Response

We thank the Reviewer for identifying and raising this citation error.

The reference has been replaced with the correct one:

19. Ulloque-Badaracco JR, Alarcón-Braga EA, Hernandez-Bustamante EA, et al. Acceptance towards Monkeypox Vaccination: A Systematic Review and Meta-Analysis. *Pathogens*. 2022;11(11):1248. doi:10.3390/pathogens11111248

20. Lounis M, Riad A. Monkeypox (MPOX)-Related Knowledge and Vaccination Hesitancy in Non-Endemic Countries: Concise Literature Review. *Vaccines*. 2023;11(2):229. doi:10.3390/vaccines11020229

21. León-Figueroa DA, Barboza JJ, Valladares-Garrido MJ, Sah R, Rodriguez-Morales AJ. Prevalence of intentions to receive monkeypox vaccine. A systematic review and meta-analysis. Published online 2023. Accessed November 18, 2023. <https://www.researchsquare.com/article/rs-3387241/latest>

Comment (2)

2. In line 161-162 the authors indicate that they used a Freeman-tukey double arcsine transformation to include studies with proportions close to or equal to 1, however, in these cases the correct thing to do is to use a continuity correction and not a variance stabilizer.

Response

We thank the Reviewer for this suggestion. Continuity correction is used to avoid the exclusion of studies with a prevalence of zero. Of note, none of our included studies reported a prevalence of zero. According to the references cited below, the use of variance stabilizers such as the Freeman-tukey double arcsine transformation is recommended when proportions are close to 0 or 1. This is because when the proportions are close to the margins (0 and 1), the distribution of the proportions may not be approximately normally distributed. Thus, to make the normal distribution assumptions more applicable in such scenarios, transformations such as the Freeman-tukey double arcsine transformation or logit transformation are needed to stabilize the variance. We have added a reference to that in our methods section, and more references are given below.

1. Barendregt, J. J., Doi, S. A., Lee, Y. Y., Norman, R. E., & Vos, T. (2013). Meta-analysis of prevalence. *J epidemiol community health*, 67(11), 974-978. <http://dx.doi.org/10.1136/jech-2013-203104>
2. Lin, L., & Xu, C. (2020). Arcsine-based transformations for meta-analysis of proportions: Pros, cons, and alternatives. *Health Science Reports*, 3(3), e178. <https://doi.org/10.1002/hsr2.178>
3. Wei, J. J., Lin, E. X., Shi, J. D., Yang, K., Hu, Z. L., Zeng, X. T., & Tong, T. J. (2021). Meta-analysis with zero-event studies: a comparative study with application to COVID-19 data. *Military Medical Research*, 8(1), 1-11.
4. Lin, L., & Xu, C. (2020). Arcsine-based transformations for meta-analysis of proportions: Pros, cons, and alternatives. *Health Science Reports*, 3(3), e178.
5. Nyaga, V. N., Arbyn, M., & Aerts, M. (2014). Metaprop: a Stata command to perform meta-analysis of binomial data. *Archives of Public Health*, 72, 1-10.

Comment (3)

3. In lines 174 to 176, the authors indicate having used Egger's test and Begg's test in search of evidence of publication bias. However, this is not a practice recommended by the scientific literature regarding meta-analyses of proportions.

(<https://bmcmedresmethodol.biomedcentral.com/articles/10.1186/s12874-021-01381-z>)

(<https://pubmed.ncbi.nlm.nih.gov/24794697/>).

Response

We thank the Reviewer for this important suggestion. We are aware of the limitations of these two tests (Egger's test and Begg's plot) we used to assess for publication bias. However, even the low accuracy for funnel plots, which has the most limitations has been shown to be in proportional studies that have low outcomes even according to the references given by the reviewer ((<https://pubmed.ncbi.nlm.nih.gov/24794697/>)). Furthermore, none of the references provided by the Reviewer outrightly invalidate these tests. We are also yet to know of a "consensus" means to assess for publication bias in proportional studies.

However, we've explored an additional method, the use of a DOI plot as recommended by more recent evidence (Furuya-Kanamori, L., Barendregt, J. J., & Doi, S. A. (2018). A new improved graphical and quantitative method for detecting bias in meta-analysis. *JBIC Evidence Implementation*, 16(4), 195-203. DOI: 10.1097/XEB.0000000000000141). We've described this in the Methods section (page 10, lines 238-244) and provided the results in our Results section (page 14, lines 325-333). Please see Supplementary Figures 15 to 28 and Table 5 for the analysis results.

Comment (4)

4. In Table 1, the article by Sagy et al. [Citation number 60]), is a retrospective cohort, however, when reviewing the section of the manuscript "Critical appraisal (quality assessment) of included studies" and observing the quality assessment scoring table, the authors only used the Newcastle-Ottawa Scale (NOS) for Cross-sectional studies. Please, it is necessary to correct this and evaluate the article in question using a specific scale for cohort studies.

Response

We thank the Reviewer for this important comment. We've made the correction and used the Newcastle Ottawa Scale for cohort studies to score this study and the other cohort studies added during our update. This can be found in Table 3b of our Supplementary Information.

Comment (5)

5. In the discussion, the authors should compare the results of their study with previous systematic reviews on vaccination acceptance for mpox.

Response

We thank the Reviewer for this important suggestion. A mention is made of the previous reviews, all of which reported only one (intention) of the three outcomes we considered (see page 25 lines 607 to 609).

REVIEWER 3 COMMENTS

I would thank the authors for this interesting work. However, after reviewing, I found some concerns that bothered me:

Comment (1)

1. The first concern is related to the fact that the authors included some studies conducted in 2020 which could affect the results. In fact, the COVID-19 pandemic has given a new vision of this phenomenon. Also, including 2 studies among 42 could affect the results.

Response

We thank the Reviewer for this important suggestion and agree with the point raised. The COVID-19 pandemic has profoundly affected health behavior especially vaccine confidence around the world according to

a plethora of published evidence. We have excluded the two studies conducted before the COVID-19 pandemic (doi:10.1016/j.vaccine.2020.08.034; doi:10.1016/j.cegh.2020.04.024), both of which were from Indonesia by the same author among healthcare workers.

Comment (2)

2. The second concern is related to the manner of writing. In fact, the authors used nearly the same expression (among. ..) in almost all the results.

Response

We thank the Reviewer for this important suggestion. We have made all necessary corrections within the manuscript and rephrased appropriately.

Comment (3)

3. Some spelling errors are also observed (exp: line 16- 17: EUR)

Response

We thank the Reviewer for this correction. These issues have also been addressed. The abbreviations for the names of the six global regions were in accordance with their usage by the WHO.

Comment (4)

3. Some spelling errors are also observed (exp: line 16- 17: EUR

17.9%; AMR 13.3%; and AFR 5.0%." add the brackets;

Response

We thank the Reviewer for this important suggestion. We have addressed this. Moreover, the writing was in accordance with the journal's formatting guidelines.

Comment (5)

line 17-18: Acceptance and uptake were 72.0% and 30.1% among the LGBTQI+ community, and healthcare workers (59.8%)" Correct, please.

Response

We thank the Reviewer for this important suggestion. This has also been corrected

Comment (6)

Line 159: delete p3

Response

We thank the Reviewer for the correction. This has also been corrected as suggested.

Comment (7)

Line 224-239: the term "by both Begg's test and Egger's test" is repeated multiple times. try to reformulate. Also "p" value should be in lowercase (in all the manuscripts).

Response

We thank the Reviewer for this important suggestion. We used both tests (and no added Doi plots tests) to evaluate for publication bias for all our outcomes which was why we reported each for the individual outcomes. The p-values have been corrected and written in lowercase.

Comment (8)

The same remark for lines 243-252 and 265-275 (then is repeated multiple times). Try to reformulate, please.

Response

We thank the Reviewer for this important suggestion. We have made appropriate corrections in the stated places and other all applicable places.

Comment (9)

Also, you do not need to repeat "meta-analysis" in each tile, you can use just: "prevalence...".

Response

We thank the Reviewer for this important suggestion. The word “meta-analysis” has now been replaced with “prevalence” in the subtitles as recommended.

Comment (10)

At last, I suggest to reformulate the title

Response

We thank the Reviewer for this important suggestion. The title of the study has now been changed to: “Mapping global prevalence and correlates of mpox vaccine acceptance and uptake in key populations: a systematic review and meta-analysis”

REVIEWERS' COMMENTS:

Reviewer #1 (Remarks to the Author):

Accept

Reviewer #2 (Remarks to the Author):

No further comments

Reviewer #3 (Remarks to the Author):

I would like to thank the authors for their efforts to improve the quality of this manuscript. Even though they answered most of my comments, they failed to improve the quality of the writing, which is not suitable for publication.

In the results, they repeated the same sentences and paragraphs in (ie: the overall global prevalence, stratified by.....)(the comment for lines 279-319 was not taken into consideration). Also, for line 465-515, the description is very simplistic and no efforts of synthesis was done (this part was better in the first vesion of the manuscript even it was not very well done)

Response to Comments by the Editor and Reviewers

General Response;

We thank the Editor for accepting our work “in principle” and for also granting us the opportunity to improve our manuscript for the final time. In this version, we have addressed the pending concern about paraphrasing certain aspects of Results section. We have also restructured the manuscript according to the journal’s formatting guidelines as contained in the special checklist sent to us. We have attached table/figure updates and other relevant files as necessary.

Editor’s Comments

Reviewer 3 comments

I would like to thank the authors for their efforts to improve the quality of this manuscript. Even though they answered most of my comments, they failed to improve the quality of the writing, which is not suitable for publication. In the results, they repeated the same sentences and paragraphs in (ie: the overall global prevalence, stratified by.....)(the comment for lines 279-319 was not taken into consideration). Also, for line 465-515, the description is very simplistic and no efforts of synthesis was done (this part was better in the first version of the manuscript even it was not very well done)

Response

We thank the review for this comment. We followed the pattern in narrating our findings in the Results section based on our belief that such way provides a more comprehensiveness of the manuscript. However, we’ve now paraphrased relevant portions of the areas concerned by the Reviewer. Please see lines 280 – 490 of the manuscript with tracked changes.